

# Fredholm determinants, full counting statistics and Loschmidt echo for domain wall profiles in one-dimensional free fermionic chains

**Oleksandr Gamayun[1★], Oleg Lychkovskiy[2,3] and Jean-Sébastien Caux[1]**

**1** Institute of Physics and Institute for Theoretical Physics, University of Amsterdam, Postbus 94485, 1090 GL Amsterdam, The Netherlands
**2** Skolkovo Institute of Science and Technology, Bolshoy Boulevard 30, bld. 1, Moscow 121205, Russia
**3** Steklov Mathematical Institute of Russian Academy of Sciences, 8 Gubkina St., Moscow 119991, Russia

★ o.gamayun@uva.nl

## Abstract

We consider an integrable system of two one-dimensional fermionic chains connected by a link. The hopping constant at the link can be different from that in the bulk. Starting from an initial state in which the left chain is populated while the right is empty, we present time-dependent full counting statistics and the Loschmidt echo in terms of Fredholm determinants. Using this exact representation, we compute the above quantities as well as the current through the link, the shot noise and the entanglement entropy in the large time limit. We find that the physics is strongly affected by the value of the hopping constant at the link. If it is smaller than the hopping constant in the bulk, then a local steady state is established at the link, while in the opposite case all physical quantities studied experience persistent oscillations. In the latter case the frequency of the oscillations is determined by the energy of the bound state and, for the Loschmidt echo, by the bias of chemical potentials.


# 1 Introduction

Quantum non-equilibrium dynamics of one-dimensional systems is a fascinating subject that nowadays attracts a lot of attention. This happens not only because of the tremendous progress in experiments but also due to the development of novel theoretical ways to understand dynamical properties in terms of universal general concepts. This is particularly true for integrable one-dimensional systems. A natural way to test such concepts is to consider evolution of the physical system after some parameters of the Hamiltonian have been rapidly changed — this is the so-called quantum quench setup [1–4]. After a long time a quenched integrable system equilibrates to a steady state that can be described in the quench action formalism [5,6] or by a complete generalized Gibbs ensemble (GGE) [7–14].

Another possibility for a quench is to consider a highly spatially inhomogeneous initial state. After long time evolution from such a state a non-vanishing flow can still be present and the system relaxes to a *non-equilibrium steady state*. This setup is the most relevant for transport properties and can be treated within the *generalized hydrodynamics* approach [15–17]. The essence of this approach lies in the existence of the Euler scale, which roughly determines sizes of cells in the system where the local conserved quantities (number of particles, momentum, energy etc) change slowly enough, in both space and time. Each such cell can be described within its local GGE, while changes from cell to cell are governed by continuity equations [18].

The generalized hydrodynamics approach sheds new light on old questions which first appeared in the theory of noise in transport in mesoscopic systems [19,20]. A central object in

this theory is the *full-counting statistics* (FCS), which is the generating function for moments of the transmitted charge. Remarkably, it was demonstrated that the leading contribution of the long-time behavior of FCS is universal and can be described by the Levitov-Lesovik formula [21–23]. From the generalized hydrodynamics point of view, this formula corresponds to the ballistic transport and can be obtained from the large deviation theory for the statistics of non-equilibrium currents [24].

Here we present a detailed study of the non-equilibrium dynamics of noninteracting lattice fermions in a system of two one-dimensional lattices connected by a link. Initially, only one of the lattices is populated with fermions up to some chemical potential. The link that connects filled and empty parts has a hopping amplitude that can differ from that in the bulk, in which case the link can be regarded as a defect in the otherwise homogeneous linear lattice. In what follows, the case when the hopping constants at the link and in the bulk are equal is referred to as the "absence of the defect". To characterize the flow of fermions through the link we compute FCS employing the form-factor expansion and present the answers in the thermodynamic limit in terms of Fredholm determinants. Contrary to the traditional approaches that present FCS in form of functional determinants [21, 22, 25], our expression is explicit [26], easily generalizable to any Gaussian initial state and does not require any additional regularization.

We use the exact expression of the FCS to extract the first two cumulants which give the current through the defect and variance of the particle number in one of the lattices. Also, the knowledge of the analytic form of the FCS allows us to compute time dependence of the bipartite entanglement entropy following the approach developed in Refs. [27–30].

In the large time limit we manage to represent the kernel of the Fredholm determinant in the expression for FCS as a generalized sine-kernel [31], which allows us to find leading asymptotic (the Levitov-Lesovik formula) and subleading logarithmic corrections. The presence of bound state provides oscillating long time behavior of the FCS, which results in an alternating non-vanishing current though the defect, the variance and the entropy rate. The frequency of these oscillations is given by the energy of the bound state.

Additionally, we calculate the Loschmidt echo (the return amplitude), which is also expressible as a Fredholm determinant. The presence of the bound state also induces oscillations in the asymptotic behavior, however, the corresponding frequency is not determined solely by the energy of the bound state but also depends on the filling of the initial state.

We underpin our analytical results in the thermodynamic limit by numerical calculations for finite size systems.

There is a vast body of literature on inhomogeneous quenches, and we verify that our formulae reproduce various previously reported results. In particular, the fully-filled half-chain in the absence of the defect in our setting corresponds to the problem with the initial domain-wall profile in the XX spin chain. In this case FCS was considered in Ref. [32, 33], where the authors managed to connect it to the random-matrix theory and also express as a Fredholm determinant of the Bessel type kernel. We verify that the proper change of basis provides a direct transformation between their and our kernels. We reproduce results on the average number of particles and its variance [34–36] (see also a numerical study in [37]) and on the Loschmidt echo [36] in the absence of the defect, as well as on the current for the infinite temperature initial state in the presence of the defect [38]. Known asymptotic results for the entanglement entropy [39–41] are also reproduced. When presenting our results in the next section, we compare them to the prior work in detail. We remark that inhomogeneous quenches in one-dimensional systems are being addressed in a wider context of interacting integrable models such as the XXZ chain [42, 43] (the result of the latter reference for the Loschmidt echo is consistent with our result in the appropriate limit) and nonintegrable models such as the interacting resonant level model [44–47], nonintegrable Ising spin chain [48] and various models of the molecular and superconducting junctions [49–52].

The paper is organized as follows. In Sec. (2) we introduce the model under consideration and list the main results. In Sec. 3 the derivation of the Fredholm determinant representation of FCS is presented, and in Sec. 4 we derive the large time limit of FCS. Secs. 5, 6 and 7 contain the derivation of the result for the cumulants (current and shot noise), entanglement entropy and Loschmidt echo, respectively, as well as a number of additional results and observations. The discussion can be found in Sec. 8. The appendix contains technical results and provides some reference materials for the Bessel functions.

## 2 Model and results

### 2.1 Model

We consider a system consisting of two one-dimensional fermionic chains connected by a link. The system is described by the tight-binding quadratic Hamiltonian

$$H = -\frac{1}{2}\sum_{j=-N+1}^{-1}\left(c_j^+ c_{j+1} + c_{j+1}^+ c_j\right) - \frac{1}{2}\sum_{j=0}^{N-2}\left(b_j^+ b_{j+1} + b_{j+1}^+ b_j\right) - \frac{\varepsilon}{2}(c_0^+ b_0 + b_0^+ c_0). \tag{1}$$

Here the first and the second term describe respectively the left and the right chain, each containing $N$ sites, while the third term describes the link between the chains. We choose to use different notations, $c_j$ and $b_j$, for fermionic operators in the left and right chains, respectively. Note that $j$ runs from $-(N-1)$ to 0 for $c_j$ and from 0 to $(N-1)$ for $b_j$. Planck's and Boltzmann's constants are set to unity throughout the paper.

The sign of $\varepsilon$ in the above Hamiltonian can always be changed by redefining $b_j \to -b_j$. For this reason we consider

$$\varepsilon \geq 0, \tag{2}$$

throughout the paper without loss of generality.

Initially the chains are disjointed (i.e. $\varepsilon = 0$). The left chain is prepared either in its ground state (i.e. at zero temperature) with fixed number of particles, $N_f$, or in the thermal grand canonical state with inverse temperature $\beta$ and chemical potential $\mu$. In the latter case $N_f$ refers to the average number of fermions.[1] We introduce the filling factor $\Phi$ proportional to the particle density,

$$\Phi \equiv \pi N_f / N. \tag{3}$$

We refer to the cases $\Phi = \pi$ and $\Phi < \pi$ as full filling and partial filling, respectively. The right chain is initially empty. The out-of-equilibrium evolution starts at $t = 0$, when $\varepsilon$ is tuned to a finite value and the fermions begin to flow to the right chain.

We consider the thermodynamic limit, i.e. the limit of $N \to \infty$ with the particle density, temperature and chemical potential of the initial state kept fixed.

In the thermodynamic limit the single-particle eigenenergies $E$ of the Hamiltonian (1) form a continuous band,

$$E \in [-1, 1]. \tag{4}$$

In addition, if $\varepsilon > 1$, two bound states appear with energies outside this band. These energies are equal to $\pm\mathcal{E}$, where

$$\mathcal{E} = (\varepsilon + 1/\varepsilon)/2. \tag{5}$$

---

[1] In Secs. 4 and 6 we also present some results for more general, nonequilibrium initial conditions for the left chain. See also a remark in footnote 2.

## 2.2 Full counting statistics

### 2.2.1 Exact expression

Our main object of interest is the thermodynamic limit of the full counting statistics (FCS), defined as

$$\chi(\lambda) \equiv \langle e^{\lambda N_R(t)} \rangle, \qquad N_R(t) = e^{itH} \left( \sum_{j=0}^{N-1} b_j^+ b_j \right) e^{-itH}. \tag{6}$$

Here and in what follows $\langle ... \rangle$ denotes the quantum mechanical average over the initial state. $N_R(t)$ is the operator of the number of fermions in the right chain in the Heisenberg picture. It should be kept in mind that the FCS (6) depends on the parameter $\lambda$, time $t$, inverse temperature $\beta$ and chemical potential $\mu$ (or, alternatively, the filling factor $\Phi$ for zero temperature), although only $\lambda$ enters the notation $\chi(\lambda)$ explicitly.

The direct physical meaning of FCS is that it is a generating function for the probability $P_n(t)$ to find exactly $n$ fermions in the right chain at time $t$,

$$P_n(t) = \int_0^{2\pi} e^{-i\lambda n} \chi(i\lambda) \frac{d\lambda}{2\pi}. \tag{7}$$

As a consequence, FCS completely describes transport properties through the link, as is discussed in detail in what follows.

There are two approaches to computing FCS in the thermodynamic limit. The first one is to consider an infinite system from the outset, and compute the correlation matrix $C_{ij} = \langle b_i^+(t) b_j(t) \rangle$, where $b_j(t)$ is the fermionic operator in the Heisenberg picture. Then the FCS is given by the determinant of an infinite matrix, $\chi(\lambda) = \det_{i,j \in \mathbb{Z}_+} \left( \delta_{ij} + (e^\lambda - 1) C_{ij} \right)$ [27,32,53]. This formula requires regularization, which introduces certain difficulties, in particular for the asymptotic analysis at large times.

We adopt a different approach. We consider a form-factor expansion in the finite system, perform exact summation (in a way similar to that in Ref. [54]) and only take the thermodynamic limit in the final expression. As a result, we express the FCS as a Fredholm determinant,

$$\chi(\lambda) = \det\left(1 + (e^\lambda - 1)D\right), \tag{8}$$

where $D$ is a linear integral operator acting on $L_2([-1,1])$ with the kernel

$$D(E, E') = \sqrt{\rho(E)} K(E, E') \sqrt{\rho(E')}, \quad E, E' \in [-1, 1]. \tag{9}$$

Here

$$\rho(E) = \frac{1}{e^{\beta(E-\mu)} + 1} \tag{10}$$

is the single-particle energy density of the initial state, i.e. the probability to find a fermion in the single-particle eigenmode with the energy $E$,[2]

$$K(E, E') = 2 \operatorname{Re} V(E, E') + (1 + \varepsilon^2) e_+(E) e_+^*(E'), \tag{11}$$

---

[2] All results derived for the equilibrium Fermi-Dirac distribution (10) at a finite temperature are also valid for an arbitrary (non-thermal) smooth single-particle energy density of the initial state.

the star refers to the complex conjugation,

$$V(E, E') = \frac{e_+(E)e_-(E') - e_+(E')e_-(E)}{E - E'}, \tag{12}$$

$$e_-(E) = \frac{e^{-itE/2}}{\sqrt{2\pi}}\sqrt{v(E)}, \qquad e_+(E) = e_-(E)S(E), \tag{13}$$

$$S(E) = \frac{1}{2\pi}\int_{-1}^{1}\frac{d\tilde{E}}{v(\tilde{E})}T(\tilde{E})\frac{e^{it\tilde{E}} - e^{itE}}{E - \tilde{E}} + \theta(\varepsilon^2 - 1)\frac{\sqrt{\mathcal{E}^2 - 1}}{2\mathcal{E}}\left(\frac{e^{-it\mathcal{E}} - e^{itE}}{\mathcal{E} + E} + \frac{e^{it\mathcal{E}} - e^{itE}}{-\mathcal{E} + E}\right) \tag{14}$$

and

$$T(E) = \frac{v(E)^2}{\mathcal{E}^2 - E^2}, \qquad v(E) = \sqrt{1 - E^2}. \tag{15}$$

Note that the last term in Eq. (14) is present only when $\varepsilon > 1$, i.e. when there is a pair of bound states in the spectrum. Functions $v(E)$ and $T(E)$ defined by Eq. (15) are the group velocity of single-particle excitations in the bulk and the transmission coefficient at the link, respectively.

A few remarks on the notion of a Fredholm determinant are in order. A straightforward and intuitive way of understanding a Fredholm determinant $\det(1-A)$ of an integral operator $A$ defined on $L_2([-1,1])$ with the kernel $A(E, E')$, $E, E' \in [-1, 1]$, is to regard it as a limit $M \to \infty$ of the determinant of the $M \times M$ matrix $\|\delta_{ab} - A(E_a, E_b)\|$, where the set $\{E_a, a = 1, 2, \ldots, M\}$ is a uniform discretization of the interval $[-1, 1]$ and $\delta_{ab}$ is the Kronecker delta. In fact, this is precisely the way how Fredholm determinants emerge in our calculations. Another insight into the notion of the Fredholm determinant is given by a useful expansion

$$\log\det(1-A) = -\sum_{n=1}^{\infty}\frac{1}{n}\operatorname{tr}A^n. \tag{16}$$

Here trace is understood as a convolution, e.g. $\operatorname{tr}A^2 \equiv \int_{-1}^{1}dE\int_{-1}^{1}dE' A(E, E')A(E', E)$. Note that, for our purposes, an integral operator $A$ is synonymous to its kernel $A(E, E')$, and we often will not discriminate between these two notions. A mathematically rigorous theory of Fredholm determinants covering, in particular, the questions of the convergence of the relevant limits and expansions, as well as the relation to the theory of integral operators and differential equations can be found e.g. in Ref. [55, 56]. Importantly, efficient methods to numerically evaluate Fredholm determinants exist [57], which make our expression (8) and other Fredholm determinants encountered in this paper as good as any tabulated special function.

It should be noted that instead of the symmetric kernel $D(E, E') = D(E', E)$ one can use the kernel $\tilde{D}(E, E') = \rho(E)D(E, E')$. The FCS is then given by Fredholm determinant $\det\left(1 + (e^\lambda - 1)\tilde{D}\right)$ of the corresponding integral operator $\tilde{D}$. This is a straightforward consequence of the identity $\det(1 + AB) = \det(1 + BA)$ valid for arbitrary invertible matrices $A$ and $B$. This remark applies also to other Fredholm determinants with a similar structure encountered in the present paper.

We also derive an alternative representation of the FCS in terms of the Fredholm determinant in the time domain,

$$\chi(\lambda) = \det\left(1 + (e^\lambda - 1)G\right)\Big|_{L_2([0,t])}, \tag{17}$$

where the kernel $G(t', t'')$ is defined on $[0, t] \times [0, t]$ and is given by Eqs. (69), (66), (70) and (71). This representation proves to be convenient for calculating the large time asymptotics of the current and shot noise, as is discussed in what follows. The derivation of the formulae from this subsection is given in Sec. 3.

### 2.2.2 Large time asymptotics

A particularly attractive feature of the Fredholm determinant representations obtained in the present paper is that they are amenable to asymptotic analysis at large times. In particular, we find in the leading order

$$\log \chi(\lambda) \approx \log \chi^{\text{hydro}}(\lambda) = \frac{t}{2\pi} \int\limits_{-1}^{1} dE \log \big(1 + (e^\lambda - 1) T(E)\rho(E)\big), \qquad t \to \infty, \qquad (18)$$

where the $\approx$ notation indicates asymptotic equality. This leading asymptotics reproduces the essentially quasiclassical (hydrodynamic) result known as the Levitov-Lesovik formula [21–23] (see also the large deviation theory perspective in Ref. [58]). The derivation of this asymptotics is presented in Sec. 4. The key step in this derivation is to approximate $K(E, E')$ defined in Eq. (11) by the generalised sine kernel, whose asymptotics is already known from Ref. [31].

In general, FCS can be written as

$$\chi(\lambda) = \mathcal{R}\, \chi^{\text{hydro}}(\lambda), \qquad (19)$$

where the time-dependent prefactor $\mathcal{R} = \mathcal{R}(t)$ contains the subexponential terms of the asymptotic expansion. As will be seen in what follows, this subleading prefactor can be very important. It is addressed in the large time limit in Sec. 4. We present analytical arguments and verify numerically that in this limit $\mathcal{R}(t)$ is a constant for $\varepsilon < 1$ and a bounded positive oscillating function for $\varepsilon > 1$. The latter oscillations are due to the pair of bound states in the single-particle spectrum, the frequency being equal to $2\mathcal{E}$. The case $\varepsilon = 1$ turns out to be special and should be treated separately. In particular, for the case of zero temperature and full filling a representation of FCS in terms of a Fredholm determinant with the Bessel kernel was found in a recent article [33]. Since the asymptotics of such Fredholm determinant is known [59], we immediately obtain

$$\begin{aligned}
\log \mathcal{R}(t) = &\frac{c\,\lambda^2}{(2\pi)^2} \log(4t) + \\
&+ 2\log\left[ G\left(1 + \frac{i\lambda}{2\pi}\right) G\left(1 - \frac{i\lambda}{2\pi}\right) \right] + o(1), \quad \varepsilon = 1, \quad \Phi = \pi, \qquad t \to \infty, \quad (20)
\end{aligned}$$

where $G(x)$ is the Barnes G-function and $c = 1$. Observe that this is an asymptotically exact expression, in the sense that it contains all nonvanishing terms in the large time limit. We also find numerically that in the case of partial filling and zero temperature the logarithmic term in the asymptotic expansion of $\log \mathcal{R}(t)$ has the same form as in Eq. (20), but with $c = 3/2$. The non-analytic dependence of $c$ on the filling is a remarkable fact discussed in more detail in Sec. (4).

## 2.3 Current, shot noise and higher cumulants

### 2.3.1 Exact expressions

As was already mentioned, the importance of the FCS (6) is that it completely determines the transport properties through the link. Indeed, its expansion in $\lambda$ encodes all irreducible moments (cumulants) $C_k$ of the particle number operator $N_R(t)$ in the right chain:

$$\log \chi(\lambda) = \sum_{k=1}^{\infty} \frac{\lambda^k C_k}{k!}, \quad C_1 = \langle N_R(t)\rangle \text{ and } C_k = \langle (N_R(t) - \langle N_R(t)\rangle)^k \rangle \text{ for } k \geq 2. \qquad (21)$$

The cumulants $C_1$ and $C_2$ have a very clear physical meaning: $C_1$ is the average particle number in the right chain while $C_2$ is the shot noise. The current through the link is given by

$$J(t) = \frac{dC_1}{dt}. \tag{22}$$

Eq. (21), in view of the expansion (16), implies that the $k$th cumulant can be expressed through traces of the $k$th and lower powers of $D$ or $G$. In particular,

$$C_1 = \operatorname{tr} D = \operatorname{tr} G, \quad C_2 = \operatorname{tr} D - \operatorname{tr} D^2 = \operatorname{tr} G - \operatorname{tr} G^2. \tag{23}$$

This way one can obtain any cumulant either from Eqs. (9)-(15) or from Eqs. (17), (69), (66), (70) and (71).

It should be mentioned that in the case of $\varepsilon = 1$ and full filling a simple exact formula (143) for the current is known, see Ref. [35] and the arXiv version of [36]. In Appendix A.2 we show how this formula can be obtained from our general results.

### 2.3.2 Large time asymptotics

In Sec. 5 we obtain large time asymptotics of the derivatives of the first two cumulants starting from Eq. (23) with the kernel $G$ from Eq. (17). The current through the link at large times reads

$$J(t) = J^{\text{hydro}} + \theta(\varepsilon^2 - 1) J^{\text{bound}} \sin(2t\mathcal{E}) + O(1/t), \quad t \to \infty, \tag{24}$$

where

$$J^{\text{hydro}} = \int_{-1}^{1} \frac{dE}{2\pi} \rho(E) T(E), \qquad J^{\text{bound}} = \frac{(\varepsilon^2 - 1)^2}{2\varepsilon} \int_{-1}^{1} \frac{dE}{2\pi} \frac{\rho(E) T(E)}{v(E)}. \tag{25}$$

$J^{\text{hydro}}$ is the well-known constant hydrodynamic contribution which can be obtained within the Landauer quasiclassical approach [38, 46, 60, 61]. This contribution immediately follows from the leading asymptotics (18) of FCS. The term proportional to $J^{\text{bound}}$ is the oscillating contribution from the pair of bound states. It is not captured by the quasiclassical approximation, which highlights the limitations of the latter. Quite remarkably, this contribution, while being of the same order as $J^{\text{hydro}}$, stems from subleading terms in the asymptotic expansion of FCS.

In the cases of zero and infinite temperatures the integrations in $J^{\text{hydro}}$ and $J^{\text{bound}}$ can be performed analytically resulting in Eq. (95) for $\beta^{-1} = 0$ and (96) for $\beta = 0$. The result (24), (96) for the infinite temperature initial state was previously obtained in Ref. [38]. It should be noted that apparently similar oscillations have recently been discovered in a nonintegrable Ising spin chain [48]. Remarkably, there the oscillations occur even in the absence of the defect and are attributed to the emergence of a bound quasiparticle at the kink of the density profile [48].

We also compute the large time behavior of the shot noise, which has a similar structure. Its time derivative reads

$$\frac{dC_2}{dt} = \int_{-1}^{1} \frac{dE}{2\pi} \rho T(1 - \rho T) + \theta(\varepsilon^2 - 1) \frac{(\varepsilon^2 - 1)^2}{2\varepsilon} B(t) + O(1/t), \quad t \to \infty, \tag{26}$$

where $B(t)$ is an oscillating function explicitly given by Eq. (107). Note that for the case of $\varepsilon = 1$ and zero temperature the above leading asymptotics of $dC_2/dt$ vanishes, and $C_2 \sim \log t$ at large times. Under the additional assumption of full filling the asymptotics of $C_2$ was calculated in Ref. [34], see Eq. (108).

### 2.4 Entanglement entropy

#### 2.4.1 Exact expressions

Entanglement entropy is an important characteristics of the equilibration process. For our setup, we define it as

$$S(t) \equiv -\operatorname{tr}\hat{\rho}_R(t) \log \hat{\rho}_R(t), \tag{27}$$

where $\hat{\rho}_R(t) = \operatorname{tr}_L \hat{\rho}(t)$ is the time-dependent reduced density matrix of the right chain obtained from the density matrix $\hat{\rho}(t) = e^{-iHt}\hat{\rho}(0)e^{iHt}$ of the full closed system by taking the partial trace $\operatorname{tr}_L$ over the left chain. Many-body density matrices $\hat{\rho}_R(t)$ and $\hat{\rho}(t)$ should not be confused with the single-particle energy density $\rho(E)$ given by Eq. (10).

The entanglement entropy of systems of noninteracting fermions can be related to FCS, as was demonstrated in refs. [27–30]. We express this relation in a simple and convenient form as

$$S(t) = \frac{1}{4} \int\limits_{-\infty}^{\infty} \frac{\log \chi(\lambda)}{\sinh^2(\lambda/2)} d\lambda, \tag{28}$$

where the integral at $\lambda = 0$ should be treated in the principal value sense.

#### 2.4.2 Large time asymptotics

The leading asymptotics for the entanglement entropy follows immediately from Eqs. (28) and (18),

$$S(t) \approx t \int\limits_{-1}^{1} \frac{dE}{2\pi} \Big( -\rho\, T \log(\rho\, T) - (1 - \rho\, T E)) \log(1 - \rho\, T) \Big), \qquad t \to \infty, \tag{29}$$

where we abbreviate the argument $E$ in $\rho(E)$ and $T(E)$. The zero temperature version of this formula can be found in Ref. [40]. The linear growth with time is generic for one-dimensional systems [62, 63].

Observe, however, that for the case of $\varepsilon = 1$ (thus, $T(E) = 1$) and zero temperature the integrand in Eq. (29) vanishes, and one needs to account for subleading terms (20) in the asymptotic expansion for the FCS. This results in

$$S(t) = \frac{c}{6} \log t + O(1), \quad t \to \infty, \quad \varepsilon = 1, \quad \beta^{-1} = 0, \tag{30}$$

with $c = 1$ for the fully-filled state (as was found in [39, 40], see also [41]) and $c = 3/2$ for the partially filled state. The logarithmic growth of entanglement in this case is consistent with general predictions of conformal field theory [40, 53, 64].

### 2.5 Return amplitude and the Loschmidt echo

#### 2.5.1 Exact expressions

Yet another important characteristic of many-body dynamics is the Loschmidt echo $\mathcal{L}(t)$, which is the probability to find the system at time $t$ in exactly the same many-body state as it was initially prepared in (at $t = 0$). The Loschmidt echo is obtained by squaring the modulus of the return amplitude $\mathcal{A}(t)$:

$$\mathcal{L}(t) = |\mathcal{A}(t)|^2, \quad \mathcal{A}(t) \equiv \langle e^{itH_0} e^{-itH} \rangle, \tag{31}$$

where $H_0 = H|_{\varepsilon=0}$ is the Hamiltonian of two disjoined chains. We find a Fredholm determinant representation for the return amplitude, which reads

$$\mathcal{A}(t) = \det(1 - W), \tag{32}$$

where the kernel $W(E, E')$ is given by

$$W(E, E') = (1 + \varepsilon^2)\sqrt{\rho(E)}V(E, E')\sqrt{\rho(E')}, \quad E, E' \in [-1, 1], \tag{33}$$

with $V(E, E')$ defined in Eq. (12).

### 2.5.2 Large time asymptotics

For $\varepsilon \neq 1$ we find in the leading order

$$\log \mathcal{A}(t) \approx \frac{t}{2\pi} \int_{-1}^{1} dE \log\left(1 - (1 + \varepsilon^2) v \, \mathcal{Z}^* \rho\right), \qquad \varepsilon \neq 1, \qquad t \to \infty, \tag{34}$$

where $\mathcal{Z}(E)$ is defined in Eq. (94). Analogously to FCS, the asymptotic expansion of $\mathcal{A}$ contains a subleading prefactor, which is oscillating in the presence of the pair of bound states (i.e. for $\varepsilon > 1$). However, in contrast to FCS, the frequency of oscillation depends not only on the energy of the bound state, but also on the filling.

The case $\varepsilon = 1$ is again special. The asymptotics (34) is not applicable in this case. Remarkably, in the case of $\varepsilon = 1$ and full filling a simple exact formula for $\mathcal{A}(t)$ was found in Ref. [36]. We rederive this formula from our general expression (32) and discuss it in a wider context in Sec. 7.

## 3 Full counting statistic: Fredholm determinant representation

In this section we outline the derivation of the Fredholm determinant representations (8) and (17) for FCS.

We start from the description of spectral properties of the Hamiltonian (1). The energy of a single-particle excitation can be parametrized by a complex number $z$ as

$$E = -(z + 1/z)/2. \tag{35}$$

Solving the single-particle eigenproblem amounts to solving the polynomial equation

$$(z^{2N+2} - 1)^2 = \varepsilon^2 z^2 (z^{2N} - 1)^2, \tag{36}$$

which can be rewritten as

$$z^{2N+2} = \frac{1 - (\pm\varepsilon z)}{1 - (\pm\varepsilon/z)}. \tag{37}$$

As can be expected, the spectrum decouples into two sets. These sets correspond to eigenfunctions which are either symmetric or antisymmetric under the reflection $b_j \leftrightarrow c_{-j}$ (see Eq. (40) below). We employ a further parametrization, $z = e^{i\phi^\pm}$ and rewrite the spectral condition (37) as

$$g(\phi^\pm) = 0 \qquad \text{with} \qquad g(\phi^\sigma) \equiv \frac{\varepsilon\cos(\phi) - \sigma}{\varepsilon\sin\phi} - \cot[\phi(N + 1)], \tag{38}$$

where $\sigma = \pm 1$ refers to the parity of the corresponding eigenfunction. In what follows we sometimes abbreviate the index $\pm$ in $\phi^\pm$, i.e. write $\phi$ instead of $\phi^\pm$. This is done mostly in

cases where $\phi^+$ and $\phi^-$ are treated on an equal footing. The dispersion relation in terms of $\phi$ reads

$$E = -\cos\phi. \tag{39}$$

The normalized single-particle eigenstates can be written as

$$|\phi^{\pm}\rangle = \frac{1}{\sqrt{\mathcal{N}(\phi^{\pm})}} \sum_{j=0}^{N-1} \sin\left[\phi^{\pm}(N-j)\right](c_{-j}^{+} \pm b_{j}^{+})|0\rangle. \tag{40}$$

Here we choose to label the eigenstates by the solutions $\phi^{\pm}$ of the spectral condition (38), $|0\rangle$ is the vacuum state annihilated by all $c_{-j}$ and $b_j$, and the normalization constant $\mathcal{N}(\phi^{\pm})$ can be conveniently presented in terms of the derivative $g'$ of the function $g(\varphi^{\pm})$,

$$\mathcal{N}(\phi) = N + 1 - \frac{1}{2}\left(1 + \frac{\sin(2N+1)\phi}{\sin\phi}\right) = g'(\phi)\sin^2[(N+1)\phi]. \tag{41}$$

Note that for $\varepsilon > 1$ there exists a pair of solutions with complex $\phi^{\pm}$, which, physically, correspond to the bound state localized at the link. These solutions are given by

$$e^{i\phi^{\pm}} = \pm 1/\varepsilon + O(e^{-N}). \tag{42}$$

To describe the initial state, we need to know the spectral properties of the Hamiltonian of the left chain detached from the right chain. Its spectrum is given by $(-\cos\varphi_n)$ with

$$\varphi_n = \frac{\pi n}{N+1}, \qquad n = 1,\dots N. \tag{43}$$

We use $\varphi_n$ to label the corresponding single-particle eigenstates:

$$|\varphi_n\rangle = e^{-i\varphi_n(N+1)}\sqrt{\frac{2}{N+1}}\sum_{j=0}^{N-1}\sin\left[\varphi_n(N-j)\right]c_{-j}^{+}|0\rangle. \tag{44}$$

Note that notations involving $\phi$ and $\varphi$ should not be confused: the former refer to the spectrum of full system, while the latter – to the spectrum of the disconnected left chain.

Now we are in a position to evaluate FCS (6). First, we do this for the zero temperature case, and then comment how to extend the result to the finite temperature case.

We start by employing Wick's theorem for group-like elements of the fermionic algebra [65] to obtain the FCS for a finite system in terms of a finite determinant,

$$\chi(\lambda) = \det X_{ab}\Big|_{a,b=1,\dots N_f},$$
$$X_{ab} = \langle\varphi_a|e^{\lambda N_R(t)}|\varphi_b\rangle = \sum_{\phi,\phi'}\langle\varphi_a|\phi\rangle\langle\phi|e^{\lambda N_R}|\phi'\rangle\langle\phi'|\varphi_b\rangle e^{-it(\cos\phi - \cos\phi')}. \tag{45}$$

Here summation over $\phi$ ($\phi'$) implies summation over all solutions of the spectral condition (38), both symmetric and antisymmetric. The form of the overlaps and matrix elements directly follows from Eqs. (40) and (44), resulting in

$$\langle\phi|\varphi\rangle = \sqrt{\frac{1}{2(N+1)\mathcal{N}(\phi)}}\frac{\sin\varphi\,\sin[(N+1)\phi]}{\cos\varphi - \cos\phi} \tag{46}$$

and

$$\langle\varphi_a|\phi\rangle\langle\phi|e^{\lambda N_R}|\phi'\rangle\langle\phi'|\varphi_b\rangle = \frac{\sin\varphi_a\sin\varphi_b(1 + \sigma\sigma'e^{\lambda})Q(\phi,\phi')}{2(N+1)g'(\phi)g'(\phi')(\cos\varphi_a - \cos\phi)(\cos\varphi_b - \cos\phi')}, \tag{47}$$

where $\sigma$ and $\sigma'$ are parities of $|\phi\rangle$ and $|\phi'\rangle$, respectively, and

$$Q(\phi,\phi') \equiv \sum_{j=0}^{N-1} \frac{\sin[(N-k)\phi_1]\sin[(N-k)\phi_2]}{\sin[(N+1)\phi_1]\sin[(N+1)\phi_2]} = \frac{\sigma - \sigma'}{2\varepsilon\sigma\sigma'(\cos\phi_1 - \cos\phi_2)}. \tag{48}$$

The above sum is evaluated as a geometric series with the help of the spectrum condition (38). To perform summation over $\phi$ and $\phi'$ it is much more convenient to consider the time derivative of $X_{ab}$, which leads to

$$\frac{dX_{ab}}{dt} = \frac{i(1-e^\lambda)\sin\varphi_a \sin\varphi_b}{2\varepsilon(N+1)}\left(F^+(\varphi_a)[F^-(\varphi_b)]^* - F^-(\varphi_a)[F^+(\varphi_b)]^*\right), \tag{49}$$

where the star means complex conjugation and functions $F^\pm$ are given by

$$F^\pm(\varphi_a) = \sum_{\phi^\pm} \frac{e^{-it\cos\phi^\pm}}{g'(\phi^\pm)(\cos\varphi_a - \cos\phi^\pm)}. \tag{50}$$

Here the sum is taken over the solutions of the spectral equation (38) with a given parity.

Now we are in a position to take the thermodynamic limit of Eq. (45). To this end we rewrite the sum (50) as a contour integral. This transformation is based on the fact that for any smooth function $f(\phi)$ the sum of $f(\phi)/g'(\phi)$ over all solutions of the non-degenerate equation $g(\phi) = 0$ can be written as

$$\sum_\phi \frac{f(\phi)}{g'(\phi)} = \oint_C \frac{d\phi}{2\pi i}\frac{f(\phi)}{g(\phi)}. \tag{51}$$

The contour $C$ encircles all spectrum points and avoids any other singularities. The special choice of $g(\phi)$ ensures $1/g(\varphi_a) = 0$ so in fact, we can encircle points $\varphi_a$ as well and deform contour into one slightly above and the other slightly below the real axis.[3] Because of the cotangent present in Eq. (38), the integrals are highly oscillatory. In the thermodynamic limit one averages these oscillations out with the help of the formula

$$\frac{1}{\pi}\int_0^\pi \frac{d\phi}{\cot\phi + z} = \frac{1}{z + i\,\mathrm{Im}z}, \qquad \mathrm{Im}z \neq 0. \tag{52}$$

Applying this procedure to the sum in Eq. (50), one gets a result which can be written, in a slight abuse of notation, as

$$F^\pm(\varphi_a) = F(\varphi_a, t, \pm\varepsilon) + O(1/N), \tag{53}$$

with

$$F(\varphi, t, \varepsilon) = -\frac{\varepsilon(1-\varepsilon\cos\varphi)e^{-it\cos\varphi}}{1+\varepsilon^2 - 2\varepsilon\cos\varphi} + \int_0^\pi \frac{d\phi}{\pi}\frac{\varepsilon^2\sin^2\phi}{1+\varepsilon^2 - 2\varepsilon\cos\phi}\frac{e^{-it\cos\phi}}{\cos\varphi - \cos\phi}$$

$$+ \theta(\varepsilon^2 - 1)\frac{\varepsilon(1-\varepsilon^2)e^{-i(\varepsilon+1/\varepsilon)t/2}}{1+\varepsilon^2 - 2\varepsilon\cos\varphi}. \tag{54}$$

Here the last term containing Heaviside theta-function emerges due to the presence of bound states, which should be treated separately in the sum (50). The integral over $\phi$ in Eq. (54)

---

[3]*cf.* summation tricks in Refs. [38,66]

is understood in the principal value sense. Observe that $\phi$, when encountered in integrals, is treated as a continuous integration variable already unrelated to the spectral condition (38).

Finally, the matrix elements are recovered from (49) taking into account the initial condition $X_{ab}\big|_{t=0} = \delta_{ab}$,

$$X_{ab} = \delta_{ab} + \frac{\pi}{N+1}(e^\lambda - 1)X(\varphi_a, \varphi_b),\tag{55}$$

where the kernel $X(\varphi, \varphi')$ reads

$$X(\varphi, \varphi') = \frac{\sin\varphi \sin\varphi'}{2\pi i\varepsilon} \int\limits_0^t d\tau \left[ F(\varphi, \tau, \varepsilon)F^*(\varphi', \tau, -\varepsilon) - F(\varphi, \tau, -\varepsilon)F^*(\varphi', \tau, \varepsilon) \right],\tag{56}$$

for arbitrary $\varphi$ and $\varphi'$. Taking into account the uniform spacing (43), we can now straightforwardly perform the thermodynamic limit to express FCS as a Fredholm determinant,

$$\chi(\lambda) = \lim_{N\to\infty} \det_{1\le a,b\le N_f} X_{ab} = \det(1 + (e^\lambda - 1)X)\Big|_{L_2([0,\Phi])}.\tag{57}$$

Observe that the range of the corresponding operator is determined by the filling fraction $\Phi$.

In fact, we can simplify the kernel (56) even further and present it as an integrable kernel [54]. To do so we introduce symmetric and antisymmetric combinations

$$S(\varphi, t) = \frac{F(\varphi, t, \varepsilon) + F(\varphi, t, -\varepsilon)}{1 + \varepsilon^2}, \qquad A(\varphi, t) = F(\varphi, t, \varepsilon) - F(\varphi, t, -\varepsilon)\tag{58}$$

and notice that they are related as

$$A(\varphi, t) = 2\varepsilon\left( i\partial_t S(\varphi, t) - \frac{e^{-it\cos\varphi}}{1 + \varepsilon^2} \right).\tag{59}$$

Thus we get

$$X(\varphi, \varphi') = \frac{\sin\varphi \sin\varphi'}{2\pi i} \int\limits_0^t d\tau \left[ S(\varphi, \tau)e^{i\tau\cos\varphi'} - S^*(\varphi', \tau)e^{-i\tau\cos\varphi} \right] + \delta X(\varphi, \varphi'),\tag{60}$$

with

$$\delta X(\varphi, \varphi') = (1 + \varepsilon^2)\sin\varphi \frac{S(\varphi, t)S^*(\varphi', t)}{2\pi}\sin\varphi'.\tag{61}$$

Further, taking into account (54), we rewrite $S(\varphi, t)$ as

$$\begin{aligned} S(\varphi, t) = \frac{1}{2} \int\limits_0^\pi \frac{d\phi}{\pi} T(\phi)\frac{e^{-it\cos\phi} - e^{-it\cos\varphi}}{\cos\varphi - \cos\phi} \\ - \theta(\varepsilon^2 - 1)\frac{\varepsilon^2 - 1}{2(\varepsilon^2 + 1)}\left( \frac{e^{-it\mathcal{E}} - e^{-it\cos\varphi}}{\mathcal{E} - \cos\varphi} + \frac{e^{it\mathcal{E}} - e^{-it\cos\varphi}}{-\mathcal{E} - \cos\varphi} \right), \end{aligned}\tag{62}$$

where

$$T(\phi) = \frac{\sin^2\phi}{\mathcal{E}^2 - \cos^2\phi}\tag{63}$$

is the transmission coefficient (this definition is consistent with Eq. (15) thanks to the dispersion relation (39)). Finally, after changing the order of integration and taking the integral

over $\tau$ in (60) we arrive at

$$X(\varphi, \varphi') = e^{it(\cos\varphi' - \cos\varphi)/2} \frac{\sin\varphi\sin\varphi'}{2\pi} \left( e^{it(\cos\varphi' + \cos\varphi)/2} \frac{S(\varphi, t) - S(\varphi', t)}{\cos\varphi - \cos\varphi'} \right.$$

$$\left. + e^{-it(\cos\varphi' + \cos\varphi)/2} \frac{S^*(\varphi, t) - S^*(\varphi', t)}{\cos\varphi - \cos\varphi'} + (1 + \varepsilon^2)S(\varphi, t)S^*(\varphi', t)e^{-it(\cos\varphi' - \cos\varphi)/2} \right). \quad (64)$$

The multiplicative factor $e^{it(\cos\varphi' - \cos\varphi)/2}$ can be dropped without affecting the value of the determinant as it is nothing but a conjugation with diagonal matrices.

The last step is to make the transformation from the kernel $X(\varphi, \varphi')$ acting in the quasi-momentum space $\varphi \in [0, \Phi]$ to the energy space. To this end one employs the dispersion relation $E(\varphi) = -\cos\varphi$ and accounts for the uniform discretization by introducing the corresponding Jacobian which equals the group velocity $v(E) = \partial E/\partial\varphi = \sqrt{1 - E^2}$. This way one gets the kernel (9) for the case of zero temperature, i.e. with $\rho(E)$ being a step function with the chemical potential reproducing the giving filling. The extension of the derivation to the case of finite temperatures (or, as a matter of fact, any finite-entropy initial state given by the single particle density $\rho(E)$) results in modifying the kernel according to $X(\varphi, \varphi') \to \sqrt{\rho(E(\varphi))}X(\varphi, \varphi')\sqrt{\rho(E(\varphi'))}$ (see book [54], or appendix A in Ref. [66]), which leads to the Fredholm determinant formula for FCS (8) with the kernel (9).

In order to obtain the representation (17) of FCS as a Fredholm determinant in the time domain, we rewrite $S(\varphi, \tau)$ as

$$S(\varphi, t) = -ie^{-it\cos\varphi} \int_0^t d\tau\, C(\tau)e^{i\tau\cos\varphi}, \quad (65)$$

with

$$C(\tau) = \int_\gamma \frac{d\phi}{2\pi} T(\phi)e^{-i\tau\cos\phi} = \int_{-1}^1 \frac{dE}{2\pi} \frac{T(E)}{v(E)} e^{-i\tau E} + \theta(\varepsilon^2 - 1)\frac{\varepsilon^2 - 1}{\varepsilon^2 + 1} \cos(\mathcal{E}\tau) \equiv \int d\mu(E)e^{-i\tau E}. \quad (66)$$

Here the contour $\gamma$ for $\varepsilon < 1$ lies on the segment $[0, \pi]$ of the real line, and for $\varepsilon > 1$ it additionally encircles in the counterclockwise direction two points given by Eq. (42) which correspond to bound states. The measure $d\mu(E)$ introduced in Eq. (66) is defined as

$$\int f(E)d\mu(E) \equiv \int_{-1}^1 f(E)\frac{T(E)}{v(E)}\frac{dE}{2\pi} + \kappa_\varepsilon f(\mathcal{E}) + \kappa_\varepsilon f(-\mathcal{E}), \qquad \kappa_\varepsilon \equiv \frac{1}{2}\frac{\varepsilon^2 - 1}{\varepsilon^2 + 1}\theta(\varepsilon^2 - 1). \quad (67)$$

Note the reflection property $C(\tau) = C(-\tau)$, which makes function $C(\tau)$ real.

From Eqs. (65) and (66) we get

$$X(\varphi, \varphi') = \frac{\sin\varphi\sin\varphi'}{2\pi} \int_0^t d^2\tau\, e^{-i\tau_1\cos\varphi} \Big[ C(\tau_1 - \tau_2) +$$

$$+ (1 + \varepsilon^2)C(\tau_1 - t)C(t - \tau_2) \Big] e^{i\tau_2\cos\varphi'}. \quad (68)$$

Finally, taking into account that $\det(1 + ACB) = \det(1 + CBA)$ for arbitrary invertible matrices $A, B$ and $C$, we rewrite FCS as a Fredholm determinant that acts in the time, Eq. (17), with the kernel

$$G(\tau_1, \tau_2) = L(\tau_1, \tau_2) + \delta L(\tau_1, \tau_2), \qquad \tau_1, \tau_2 \in [0, t], \quad (69)$$

where

$$L(\tau_1, \tau_2) = \int_0^t d\tau\, C(\tau_1 - \tau) R(\tau - \tau_2), \qquad \delta L(\tau_1, \tau_2) = (1 + \varepsilon^2) C(\tau_1 - t) L(t, \tau_2), \quad (70)$$

$$R(\tau) = \int_0^\pi \frac{d\varphi}{2\pi} \rho(\varphi) e^{i\tau \cos\varphi} \sin^2\varphi = \int_{-1}^1 \frac{dE}{2\pi} \nu(E) \rho(E) e^{-i\tau E} \equiv \int_{-1}^1 dr(E) e^{-i\tau E}, \quad (71)$$

and the measure $dr(E)$ is defined according to

$$dr = \frac{dE}{2\pi} \nu(E) \rho(E). \quad (72)$$

Observe that $R(\tau)$ encodes all information about the initial state. Note that, contrary to $C(\tau)$, $R(\tau)$ is not a priori symmetric and can have complex values. We find the Fredholm determinant representation of FCS in the time domain (17) to be well-suited for calculating cumulants.

## 4  Full counting statistics: large time behavior

In this section we address large time behavior of the FCS (8). First, we make an approximation for the kernel (11), which allows us to split it into the generalized sine-kernel and regular corrections, which can be disregarded in the leading order. Second, we obtain the leading asymptotics of $\log\chi(\lambda)$ at $t \to \infty$ using known results for the generalized sine-kernel [31, 67, 68]. The thus obtained leading asymptotics coincides with the Levitov-Lesovik formula for FCS [21–23] obtained within what can be called the hydrodynamic or semiclassical approach. Finally, we address subleading terms in the asymptotic expansion of $\log\chi(\lambda)$.

Let us start with the approximation of the integrals in (14). For any regular function $f(E)$ the principal value integral can be approximated as

$$\frac{1}{\pi i} \int_{-1}^1 \frac{dE}{E - E_0} f(E) e^{itE} \approx f(E_0) e^{itE_0}, \qquad t \to +\infty. \quad (73)$$

Therefore one can rewrite $S(E)$ in Eq. (14) as

$$S(E) = e^{itE}(p - iq) + e^{it\mathcal{E}} r^+ + e^{-it\mathcal{E}} r^-, \quad (74)$$

where

$$p(E) = \frac{\varepsilon^2 - 1}{2(\varepsilon^2 + 1)} \frac{E}{\mathcal{E}^2 - E^2}, \qquad q(E) = \frac{T(E)}{2\nu(E)}, \qquad r(E)^\pm = \frac{\varepsilon^2 - 1}{2(\varepsilon^2 + 1)} \frac{\theta(\varepsilon^2 - 1)}{E \mp \mathcal{E}}, \quad (75)$$

and the argument $E$ of the functions $p, r^\pm$ and $q$ in Eq. (74) is suppressed.

The kernel (11) can be presented as a sum of two parts $K = K_H + \delta K$,

$$K_H(E, E') = \frac{\sqrt{\nu(E)}\sqrt{\nu(E')}(q(E) + q(E'))}{\pi} \frac{\sin \frac{t(E - E')}{2}}{E - E'} \quad (76)$$

$$\delta K(E, E') = \frac{\sqrt{\nu(E)}\sqrt{\nu(E')}}{\pi} \left( \cos \frac{t(E + E' + 2\mathcal{E})}{2} \frac{r^-(E) - r^-(E')}{E - E'} \right.$$
$$\left. + \cos \frac{t(E + E' - 2\mathcal{E})}{2} \frac{r^+(E) - r^+(E')}{E - E'} + \cos \frac{t(E - E')}{2} \frac{p(E) - p(E')}{E - E'} \right)$$
$$+ \frac{1 + \varepsilon^2}{2\pi} \sqrt{\nu(E)}\sqrt{\nu(E')} S(E) S^*(E') e^{it(E' - E)/2}. \quad (77)$$

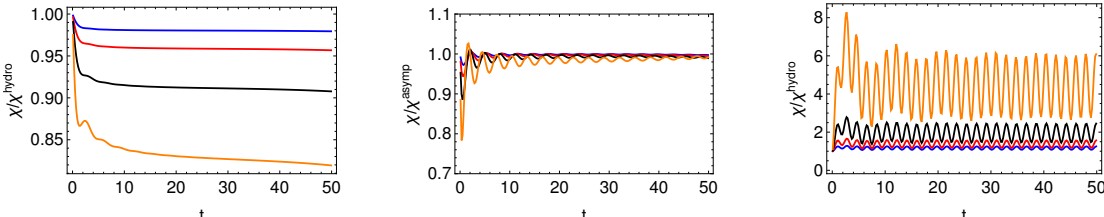

Figure 1: Left and right: the ratio $\mathcal{R}(t)$ of the exact FCS (8) to the hydrodynamic contribution (18) for the full initial filling and different hopping constants at the link, $\varepsilon = 0.3$ (left) and $\varepsilon = 3.0$ (right). These plots illustrate that at large times $\mathcal{R}(t)$ saturates for $\varepsilon < 1$ and oscillates around some average value for $\varepsilon > 1$. Center: the ratio of the exact FCS (8) to the exact large time asymptotics (20) for $\varepsilon = 1$ and the full initial filling. At small times one can see transient oscillations which get completely damped at large times. Blue, red, black and orange curves correspond to respectively $\lambda = 0.25, 0.5, 1, 2$ in all three plots.

In the large time $t \to \infty$ due to the fast oscillations we can approximate kernel $K_H$ as

$$K_H(E, E') \approx \frac{2\sqrt{v(E)}\sqrt{v(E')}q(E)}{\pi} \frac{\sin\frac{t(E-E')}{2}}{E - E'}. \tag{78}$$

Heuristically, this can be understood considering the trace expansion of the logarithm of the determinant, and taking into account that $q(E)$ behaves regularly at the edges $E = \pm 1$ of the interval $[-1, +1]$. This approximation does not affect the leading term of the expansion but might modify the overall multiplicative constant. Taking into account that $r^{\pm}(E)$ and $p(E)$ are rational functions, one can see that all terms in (77) except the last one are factorized, and the determinant is linear in each of these terms. Therefore, applying the same logic, we can disregard $\delta K$ completely. Note that the latter approximation again affects the subleading terms – most importantly, for $\varepsilon > 1$ the oscillating term $\cos(\mathcal{E}t)$ gets lost. All these assumptions being made, we arrive at

$$\chi(\lambda) \approx \det\left(1 + T(E)\rho(E)\frac{e^{\lambda} - 1}{\pi}\frac{\sin\frac{t(E-E')}{2}}{E - E'}\right), \qquad t \to \infty. \tag{79}$$

The operator acts on the full Brillouin zone $E \in [-1, 1]$. This expression was obtained in Ref. [23], see their Eq. (37).

The kernel in Eq. (79) is a generalized sine-kernel. Its asymptotic analysis can be found in Ref. [31], with the result

$$\chi(\lambda) \approx \chi^{\text{hydro}}(\lambda) = \frac{e^{-\Omega(\lambda)t}}{t^{\alpha(\lambda)}}, \qquad t \to \infty, \tag{80}$$

with

$$\Omega(\lambda) = i\int_{-1}^{1} dE\, v(E), \quad \alpha(\lambda) = v(+1)^2 + v(-1)^2,$$

$$v(E) = \frac{i}{2\pi}\log\left[1 + (e^{\lambda} - 1)T(E)\rho(E)\right]. \tag{81}$$

In fact, for $\varepsilon \neq 1$ the power law factor does not appear in Eq. (80), since $T(\pm 1) = 0$ and hence $\alpha(\lambda) = 0$. Thus we obtain the leading asymptotics (18) of $\log \chi(\lambda)$. It coincides with the prediction of the large deviation theory [58] which is essentially based on quasiclassical (or, equivalently, hydrodynamical) considerations.

We plot the ratio $\mathcal{R}(t)$ of the exact FCS (8) to the leading asymptotics (18) in Fig. (1). For $\varepsilon < 1$ the subleading contributions result in a time-independent $\mathcal{R}(t)$ at large times. For $\varepsilon > 1$ one can see that $\mathcal{R}(t)$ oscillates around some finite average value at large times, as is expected from the above derivation of the leading asymptotics.

Direct application of the formula (80) for $\varepsilon = 1$ (and thus $T(E) = 1$) is not possible due to the singular behavior of the function $q(E)$ at the edges of the spectrum $E \to \pm 1$, which makes the approximation (78) too crude.

Luckily, for $\varepsilon = 1$ there is another way to find the leading asymptotics and even the subleading contributions. Let us start from the full filling case. As demonstrated in Ref. [32], FCS in this case can be expressed as a determinant with the discrete Bessel kernel (we show in the appendix (A.2) how this representation can be obtained from our Fredholm determinant with the kernel (56)). Further, as shown in Ref. [33], the determinant with the discrete Bessel kernel can be transformed to the Fredholm determinant with the continuous Bessel kernel with the result

$$\chi(\lambda) = \det(1 + (e^\lambda - 1)\hat{K}_{\mathrm{Bes}})_{L_2[0,t^2]}, \tag{82}$$

where

$$K_{\mathrm{Bes}}(x,y) = -\frac{J_0(\sqrt{x})\sqrt{y}J_1(\sqrt{y}) - J_0(\sqrt{y})\sqrt{x}J_1(\sqrt{x})}{2(x-y)}. \tag{83}$$

The asymptotic for this kernel can be readily found within the Riemann-Hilbert problem approach [59], resulting in

$$\begin{aligned}
\log \chi(\lambda) = {} & \frac{\lambda t}{\pi} + \frac{\lambda^2}{(2\pi)^2} \log(4t) \\
& + 2\log\left[G\left(1 + \frac{i\lambda}{2\pi}\right)G\left(1 - \frac{i\lambda}{2\pi}\right)\right] + o(1), \quad \varepsilon = 1, \quad \Phi = \pi, \quad t \to \infty,
\end{aligned} \tag{84}$$

where $G(x)$ is the Barnes G-function. Note that Eq. (84) is asymptotically exact, i.e. it contains all nonvanishing in the large time limit terms of $\log \chi(\lambda)$. Comparing this result with the regular hydrodynamic prediction (80) we see that the singular behavior of the kernel at the spectrum edge does not affect the leading behavior $\Omega(\lambda)$, but affects the exponent $\alpha(\lambda)$, which is twice smaller than what could be incorrectly inferred from Eq. (81). The latter fact can also be noticed from the asymptotic result (108) for the second cumulant.

We also address the case of $\varepsilon = 1$ and the partial filling. We find numerically that the constant $\alpha(\lambda)$ is affected by the filling in a non-continuous way, depending on whether the edges of the spectrum are occupied. Let us consider the initial state such that levels with $E \in [\mu_1, \mu_2]$ are occupied (for $\mu_1 > -1$ this constitutes a non-equilibrium initial condition). We find the first two terms of the large time asymptotics,

$$\log \chi(\lambda) = \frac{\lambda t}{\pi}\frac{\mu_1 - \mu_2}{2} + \frac{z(\mu_1, \mu_2)\lambda^2}{(2\pi)^2}\log(4t) + O(1), \qquad \varepsilon = 1, \tag{85}$$

where $z(\mu_1, \mu_2) = 2$, $z(-1, \mu_2) = z(\mu_1, 1) = 3/2$ and $z(-1, 1) = 1$ for $-1 < \mu_1, \mu_2 < 1$. The observed discontinuity of the exponent $\alpha(\lambda)$ as a function of the initial state is solely due to the large time limit and is in somewhat reminiscent to the fractal structure of the Drude weight in spin chains [69].

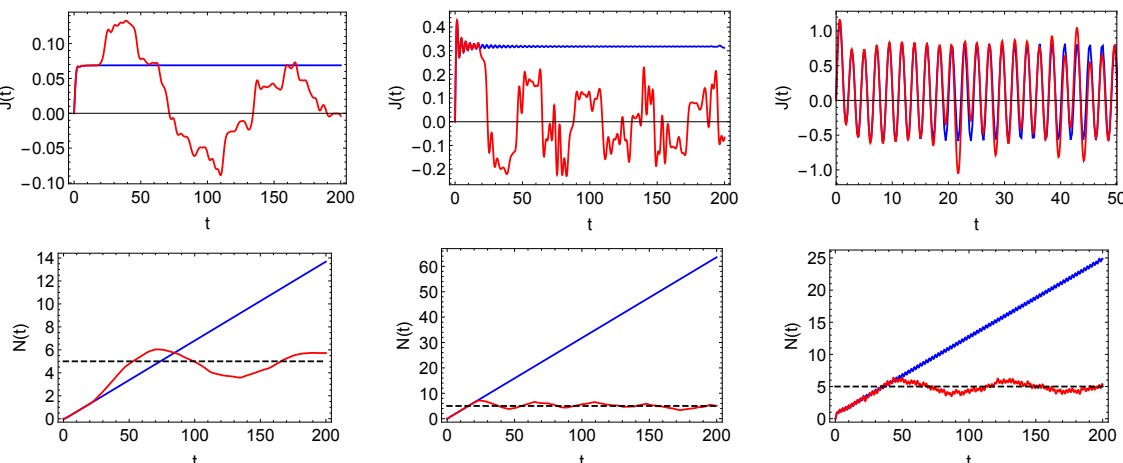

Figure 2: The current through the link (upper panels) and the number of particles in the right chain (lower panels) for $\varepsilon = 0.3$ (left), $\varepsilon = 1.0$ (center) and $\varepsilon = 2.4$ (right). Red curves correspond to numerical results for the initially fully filled left chain ($N = 10$ fermions). The current (86) in the thermodynamic limit is shown by blue solid lines. The black dashed lines in lower panels indicate the half of the initial number of particles. Notice the change of the time scale in the upper left plot.

# 5 Current and shot noise

In this section we compare the current in the thermodynamic limit to the current in the system of a finite size, and derive large time asymptotics of the current and shot noise.

In the thermodynamic limit the current through the defect can be obtained from Eq. (23) with $D(E, E')$ defined in Eqs. (9)-(15), with the result

$$J(t) = \int\limits_0^\pi \frac{d\varphi}{\pi} \rho(\varphi) \sin^2 \varphi \, \mathrm{Im} \left[ F(\varphi, t, \varepsilon) F(\varphi, t, -\varepsilon) \right], \tag{86}$$

where $F(\varphi, t, \varepsilon)$ is defined in Eq. (54). We compare this result with the current in a finite system for initial states at zero temperature. For the full filling ($\Phi = \pi$) we take $N_f = 10$ particles (see Fig. (2)) and for the filling $\Phi = \pi/3$ we take $N_f = 6$ particles (see Fig. (3)). We observe that as long as the front of the expanding Fermi gas has not bounced back from the edge of the system, the result (86) obtained in the thermodynamic limit fits the numerical result for the finite system extremely well, despite the small number of particles. Notice also that the equilibration of the number of particles in the finite system happens pretty quickly, and after a few bounces from the edges the average number of particles in the left chain is equal to half of the total number of particles.

We now turn to the analysis of the large time behavior of the current. To this end we find it convenient to use the representation (17), which gives

$$C_1 = \mathrm{tr}\, L + \mathrm{tr}\, \delta L, \tag{87}$$

where $L$ and $\delta L$ are defined in Eq. (70). Let us start with the first trace that we rewrite as

$$\mathrm{tr}\, L = \int d\mu(E_1) \int dr(E_2) \left[ \frac{2 \sin(t E_{12}/2)}{E_{12}} \right]^2, \qquad E_{12} = E_1 - E_2, \tag{88}$$

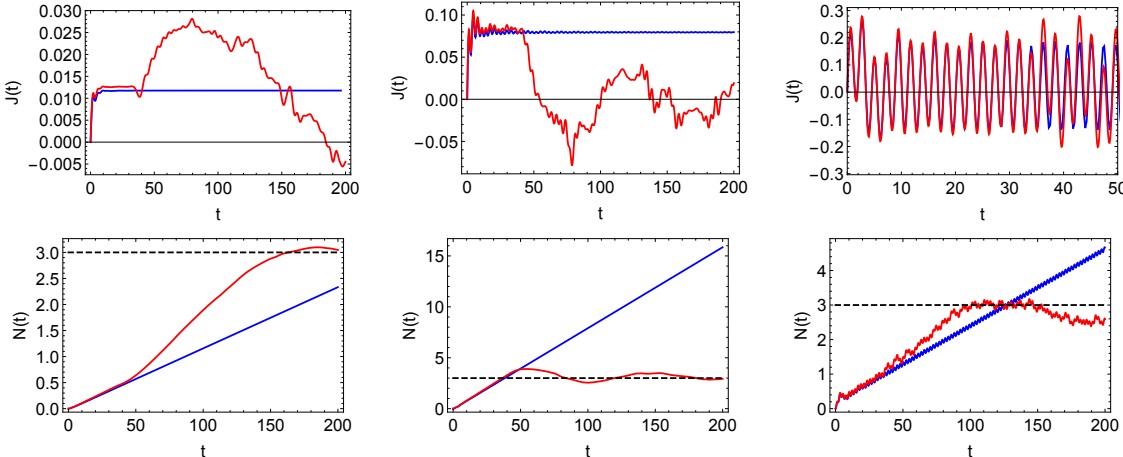

Figure 3: The current through the link (upper panels) and the number of particles in the right chain (lower panels) for $\varepsilon = 0.3$ (left), $\varepsilon = 1.0$ (center) and $\varepsilon = 2.4$ (right). Red curves correspond to numerical results for the initially partially filled left chain ($N = 6$ fermions, 18 sites in the left chain). The current (86) in the thermodynamic limit is shown by blue solid lines. The black dashed lines in lower panels indicate the half of the initial number of particles. Notice the change of the time scale in the upper left plot.

or for the derivative

$$\frac{d\,\mathrm{tr}\,L}{dt} = 2 \int d\mu(E_1) \int dr(E_2) \frac{\sin(tE_{12})}{E_{12}}. \tag{89}$$

The measures $d\mu(E)$ and $dr(E)$ are defined in Eq. (67) and (71) respectively. In the limit of large $t$ we replace $\sin(tx)/x \to \pi\delta(x)$, transforming Eq. (89) in the Landauer-Büttiker like expression

$$\frac{d\,\mathrm{tr}\,L}{dt} \approx J^{\mathrm{hydro}} \equiv \int\limits_{-1}^{1} \frac{dE}{2\pi} \rho(E)T(E), \tag{90}$$

which can be obtained via the direct expansion of the asymptotic form of the FCS (80). The oscillating behavior of the current is not captured by the Eq. (80) and is coming from the finite rank contribution

$$\frac{\mathrm{tr}\,\delta L}{1 + \varepsilon^2} = \int \prod_{i=1}^{2} d\mu(E_i) \int dr(E_3) \frac{e^{itE_{13}} - 1}{iE_{13}} \frac{e^{itE_{32}} - 1}{iE_{32}}, \tag{91}$$

where $E_{ij} \equiv E_i - E_j$. In the limit of large time the continuous contribution of the measure $d\mu(E)$ can be accounted for by the following approximation:

$$\lim_{t \to +\infty} \frac{e^{iEt} - 1}{iE} = \frac{i}{E + i0}. \tag{92}$$

Combining this approximation with the explicit contribution from the discreet part we obtain

$$\frac{\mathrm{tr}\,\delta L}{1 + \varepsilon^2} = \int\limits_{-1}^{1} \frac{dE}{2\pi} \rho(E)\nu(E)\left( |\mathcal{Z}(E)|^2 + 2\kappa_\varepsilon^2 \frac{\mathcal{E}^2 + E^2}{(\mathcal{E}^2 - E^2)^2} - 2\kappa_\varepsilon^2 \frac{\cos(2t\mathcal{E})}{\mathcal{E}^2 - E^2} \right) + O(1/t), \tag{93}$$

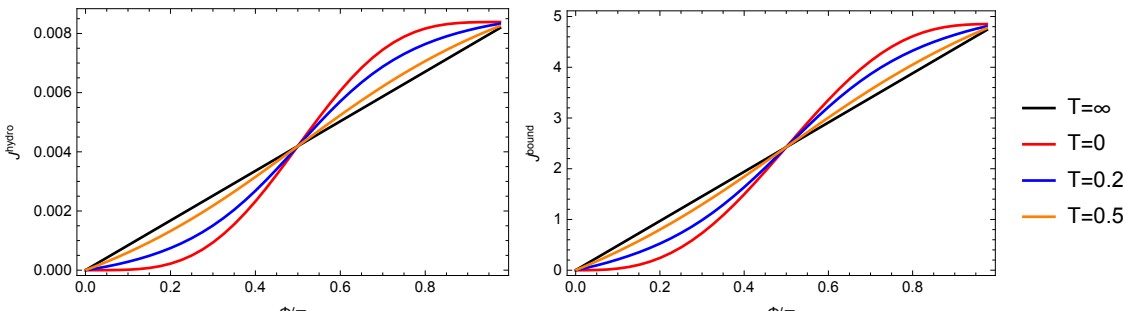

Figure 4: Amplitudes (25) of the hydrodynamic and oscillating contributions to the current (24) at large times as functions of the initial particle density in the left chain, $N_f/N = \Phi/\pi$, computed for various temperatures. The hopping constant at the link is $\varepsilon = 10$.

where $\kappa_\varepsilon$ is defined in Eq. (67) and we have introduced the function

$$\mathcal{Z}(E) = \lim_{t \to +\infty} \int_{-1}^{1} \frac{dE'}{2\pi} \frac{T(E')}{\nu(E')} \frac{e^{i(E'-E)t} - 1}{i(E'-E)} = \frac{\nu(E)/2 - i\kappa_\varepsilon E}{\mathcal{E}^2 - E^2},$$

$$|\mathcal{Z}(E)|^2 = \frac{T(E)}{\nu(E)^2} \frac{\varepsilon^2}{(1+\varepsilon^2)^2}. \tag{94}$$

The first term in Eq. (93) corresponds to the continuous part of the measure $d\mu(E)$, the second term comes from the discrete part and accounts for contribution $E_1 = \pm\mathcal{E}$, $E_2 = \pm\mathcal{E}$, while the last term accounts for $E_1 = \pm\mathcal{E}$, $E_2 = \mp\mathcal{E}$. We see that only the last terms contributes to the current, resulting in Eqs. (24), (25).

For zero temperature $J^{\text{hydro}}$ and $J^{\text{bound}}$ from Eq. (25) can be computed analytically, with the result

$$J^{\text{hydro}} = \frac{\sin^2(\Phi/2)}{\pi} + \frac{(1-\varepsilon^2)^2}{8\pi\varepsilon(1+\varepsilon^2)} \log\left[\frac{(\varepsilon-1)^2}{(\varepsilon+1)^2} \frac{1+\varepsilon^2 + 2\varepsilon\cos\Phi}{1+\varepsilon^2 - 2\varepsilon\cos\Phi}\right],$$

$$J^{\text{bound}} = \frac{(\varepsilon^2-1)^2}{8\pi\varepsilon(1+\varepsilon^2)}\left[2\Phi(1+\varepsilon^2) + (\varepsilon^2-1)\left(2\arctan\left(\frac{\varepsilon^2-1}{\varepsilon^2+1}\cot\Phi\right) - \pi\right)\right], \quad \beta^{-1} = 0. \tag{95}$$

$J^{\text{hydro}}$ and $J^{\text{bound}}$ can also be calculated analytically for the case of infinite temperature ($\rho(E)$ is independent on $E$ and equals the initial average number density of fermions in the left chain, $\Phi/\pi$). This case was addressed in Ref. [38]. We confirm their result for $J^{\text{hydro}}$ and find a minor mistake in their $J^{\text{bound}}$. Our result reads (cf. Eqs. (39) and (40) in [38])

$$J^{\text{hydro}} = \frac{\Phi}{\pi^2}\left(1 + \frac{(1-\varepsilon^2)^2}{2\varepsilon(1+\varepsilon^2)}\log\left|\frac{\varepsilon-1}{\varepsilon+1}\right|\right), \quad J^{\text{bound}} = \frac{\Phi}{\pi}\frac{(\varepsilon^2-1)^2}{2\varepsilon(1+\varepsilon^2)}, \quad \beta = 0. \tag{96}$$

Amplitudes $J^{\text{hydro}}$ and $J^{\text{bound}}$ as functions of the particle density are plotted for various temperatures in fig (4). To relate the chemical potential to the filling factor we employ the dispersion relation $E(\varphi) = -\cos\varphi$ which leads to the equation $\Phi = \int_0^\pi (1 + e^{(-\cos(\varphi) - \mu)/T})^{-1} d\varphi$.

Note that the oscillations of the current in Eq. (24) are persistent (they do not decay with time). They should be distinguished from transient oscillations observed e.g. in [46,70] which

vanish at large times. The frequency of persistent oscillations of the current is determined by the energy of bound states and does not depend on the chemical potential (filling). The frequency of the transient oscillations [46,70], in contrast, is determined solely by the chemical potential.

Now we turn to the evaluation of the large time behavior of $\frac{dC_2}{dt}$. We start from

$$C_2 = C_1 - \operatorname{tr} L^2 - 2\operatorname{tr} L\delta L - \operatorname{tr} \delta L^2, \tag{97}$$

which follows from Eqs. (23) and (69). Let us consider each term separately. The last term can be analyzed by using the result (93) for the current, since $\delta L$ is a rank one operator and, consequently, $\operatorname{tr} \delta L^2 = (\operatorname{tr} \delta L)^2$. The middle term can be presented as

$$\frac{\operatorname{tr} L\delta L}{1+\varepsilon^2} = \int \prod_{i=3}^{5} d\mu(E_i) \prod_{i=1}^{2} dr(E_i) \frac{e^{itE_{15}}-1}{iE_{15}} \frac{e^{itE_{52}}-1}{iE_{52}} \frac{e^{itE_{23}}-1}{iE_{23}} \frac{e^{itE_{41}}-1}{iE_{41}}$$

$$= \int d\mu(E_5)|\mathcal{F}(E_5)|^2, \tag{98}$$

where function $\mathcal{F}(E_5)$ involves integration over $E_2$ and $E_3$, and in the large time limit can be presented as

$$\mathcal{F}(E_5) = \int d\mu(E_3)dr(E_2) \frac{e^{itE_{52}}-1}{iE_{52}} \frac{e^{itE_{23}}-1}{iE_{23}}. \tag{99}$$

In the large time limit using approximation (92) we can simplify it as

$$\mathcal{F}(E_5) \approx F_0(E_5) + \kappa_\varepsilon e^{iE_5 t} \left[ F_-(E_5)e^{-i\mathcal{E}t} + F_+(E_5)e^{i\mathcal{E}t} \right], \tag{100}$$

in which

$$F_0(E_5) = \int_{-1}^{1} \frac{dE}{2\pi} \frac{iv(E)\rho(E)\mathcal{Z}(E)}{E_5 - E + i0}, \qquad F_\pm(E_5) = \int_{-1}^{1} \frac{dE}{2\pi} \frac{v(E)\rho(E)}{E \pm \mathcal{E}} \frac{1}{E - E_5 + i0}. \tag{101}$$

This results in

$$\frac{\operatorname{tr} L\delta L}{1+\varepsilon^2} = 2\kappa_\varepsilon^2 \operatorname{Re}\left[ e^{-2i\mathcal{E}t}\left( \left( F_0(\mathcal{E}) + F_0^*(-\mathcal{E}) + \kappa_\varepsilon(F_+(-\mathcal{E}) + F_-(\mathcal{E})) \right) F_+(\mathcal{E}) \right. \right.$$

$$\left. \left. + \int d\mu(E)F_+^*(E)F_-(E) \right) \right] + \dots, \tag{102}$$

where by dots we have denoted terms that either do not depend on time at all or vanish in the large time limit. After tedious but straightforward algebraic transformations we can simplify this expression to

$$\frac{\operatorname{tr} L\delta L}{1+\varepsilon^2} = -2\kappa_\varepsilon^2 \operatorname{Re}\left[ e^{-2i\mathcal{E}t}\left( \int \frac{dE}{2\pi} \frac{\rho T(E)}{v(E)} \int \frac{v(E)dE}{2\pi} \frac{i\mathcal{E}v(E) + \kappa_\varepsilon(3E^2 + \mathcal{E}^2)}{(\mathcal{E}^2 - E^2)^2} \right. \right.$$

$$\left. \left. + \int \frac{dE}{4\pi} \frac{\rho^2 T^2(E)}{v(E)} \right) \right] + \dots. \tag{103}$$

The first trace can be written as

$$\operatorname{tr} L^2 = \int \prod_{i=1,3} d\mu(E_i) \prod_{i=2,4} dr(E_i) \frac{2\sin(t/2E_{12})}{E_{12}} \frac{2\sin(t/2E_{23})}{E_{23}} \frac{2\sin(t/2E_{34})}{E_{34}} \frac{2\sin(t/2E_{41})}{E_{41}}. \tag{104}$$

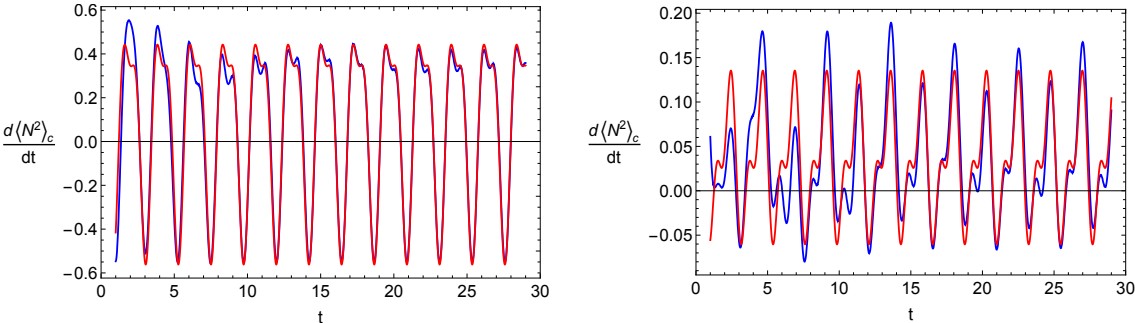

Figure 5: The rate $\frac{dC_2}{dt}$ of the second cumulant of the particle number in the right chain for $\varepsilon = 2.4$ and zero temperature initial state. The left panel corresponds to the fully filled initial state, $\Phi = \pi$, and the right one to the half-filled initial state, $\Phi = \pi/2$. The blue curve corresponds the exact expression deduced from Eqs. (21), (8), while the red one is given by the large-time asymptotic expression (26).

The continuous part of the measure is responsible for the leading contributions, which are linear in time. Namely, we approximate $\sin(tx)/x \to \pi\delta(x)$ for the parts that include $E_3$ and $E_4$ to obtain

$$\operatorname{tr} L_c^2 = \int d\mu(E_1) dr(E_2) T(E_1)\rho(E_1) \left[\frac{2\sin(tE_{12}/2)}{E_{12}}\right]^2, \qquad (105)$$

which is similar to Eq. (88) apart from the regular factors. The time-dependent contribution from the discrete part of the measure comes from the contributions $E_1 = \mathcal{E}$, $E_3 = -\mathcal{E}$ and vice versa

$$\operatorname{tr} L_d^2 = 8\kappa_\varepsilon^2 \left(\int \frac{dE}{2\pi} \frac{\rho(E)v(E)}{\mathcal{E}^2 - E^2}\right)^2 \cos(\mathcal{E}t)^2. \qquad (106)$$

Finally, combining expressions (93), (103), (106) and (105), and taking time derivative we obtain Eq. (26), where $B(t)$ describes the bound state contribution and reads

$$B(t) = \sin(2\mathcal{E}t) \int_{-1}^{1} \frac{dE}{2\pi} \rho T(1-\rho T) + \frac{(\varepsilon^2-1)^2}{2(\varepsilon^2+1)} \sin(4\mathcal{E}t) \left(\int_{-1}^{1} \frac{dE}{2\pi}\rho T\right)^2$$

$$+ 2\left(\int_{-1}^{1} \frac{dE}{2\pi}\rho T\right)\left(2\mathcal{E}\cos(2\mathcal{E}t)\int_{-1}^{1} \frac{dE}{2\pi}\frac{\rho T^2}{v^2} - \frac{1}{2}\frac{\varepsilon^2-1}{\varepsilon^2+1}\sin(2\mathcal{E}t) \times\right.$$

$$\left.\times \int_{-1}^{1} \frac{dE}{2\pi}\rho v \frac{E^2(\mathcal{E}^2-E^2)+2\mathcal{E}^2}{(\mathcal{E}^2-E^2)^2}\right). \qquad (107)$$

In Fig. (5) we compare the asymptotic expression (26) for the rate of the second cumulant with the exact expression obtained from Eqs. (21), (8).

Note that for $\varepsilon = 1$ and zero temperature the asymptotics (26) of $\frac{dC_2}{dt}$ vanishes. In this case the $O(1/t)$ term in $\frac{dC_2}{dt}$ becomes important and leads to a logarithmic term in $C_2$. For the case of $\varepsilon = 1$ and full filling the asymptotics of $C_2$ was found in Ref. [34]:

$$C_2 = \frac{1}{2\pi^2}\log t + \text{const} + o(1), \qquad \varepsilon = 1, \qquad \Phi = \pi, \quad t \to \infty. \qquad (108)$$

This result is consistent with Eq. (20).

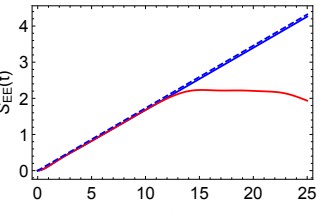 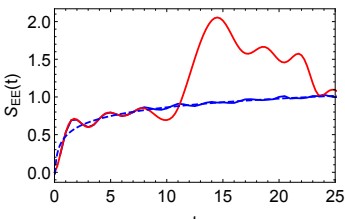 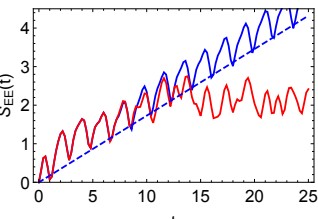

Figure 6: The growth of entanglement entropy for the fully-filled initial state and various values of the defect hopping constant, $\varepsilon = 1/3$ (left), $\varepsilon = 1$ (center) and $\varepsilon = 3$ (right). Solid blue curves stand for the exact expression (28), while the dashed blue curves stand for the asymptotics (29) (left and right panels) or (30) (middle panel). The red lines show evolution in the finite size system with $N = 5$ particles.

## 6 Entanglement entropy

Let us derive Eq. (28). The entanglement entropy (27) has the following cumulant expansion [27–30],

$$S(t) = \sum_{k>0} \frac{\alpha_k}{k!} C_k, \qquad \alpha_k = \begin{cases} (2\pi)^k |B_k|, & k \text{ even}, \\ 0, & k \text{ odd} \end{cases}, \tag{109}$$

where $B_m$ are Bernoulli numbers. We can take into account that

$$\pi^{2m} |B_{2m}| = \frac{1}{2} \int_{-\infty}^{\infty} \frac{\lambda^{2m}}{\sinh^2 \lambda} d\lambda. \tag{110}$$

Note that the r.h.s. vanishes for odd powers $2m \to 2m + 1$, so we can present

$$S(t) = \frac{1}{2} \sum_{k>0} \int_{-\infty}^{\infty} \frac{C_k (2\lambda)^k}{k!} \frac{d\lambda}{\sinh^2 \lambda} = \frac{1}{4} \int_{-\infty}^{\infty} \frac{\log \chi(\lambda)}{\sinh^2(\lambda/2)} d\lambda, \tag{111}$$

where in the last step we have used expansion (21). The integral over $\lambda$ converges quickly at infinity, while at $\lambda = 0$ it should be treated in a principal value sense.

Let us now address the asymptotic behavior of the entanglement entropy at large times. Both Eq. (29) and (30) are straightforwardly obtained from Eq. (28) and the corresponding asymptotics of FCS, Eqs. (18) and (20). The latter result featuring logarithmic growth of entanglement for $\varepsilon = 1$ at zero temperature deserves additional discussion. For an initial state already considered in Sec. 4 with occupied levels with $E \in [\mu_1, \mu_2]$ we get for Eq. (30) with $c = 1$ is for the fully-filled state [39–41], $c = 3/2$ for the partially filled state and $c = 2$ for a nonequilibrium state with $-1 < \mu_{1,2} < 1$. As shown in [40], the effective central charge $c$ in front of $\log t$ in the dynamical quench problem is identical to that in front of $\log l$ in the asymptotic expansion of the entanglement entropy of a subsystem of size $l$ in the time-independent, stationary problem. This connects our results to the thoroughly investigated subject of entanglement scaling in stationary states of free fermionic chains (see e.g. the general CFT prediction [41, 71]). In particular, the dependence of the "central charge" $c$ on the initial state resembles the observation made in Ref. [72] that this quantity is related to the scattering phase, and in principle can be made arbitrary.

We compare exact and asymptotic expressions for the entanglement entropy in the thermodynamic limit, as well as the entanglement entropy for a finite system in Fig. (6). Notice

that since the transmission coefficient $T(E)$ is invariant under the transformation $\varepsilon \to 1/\varepsilon$, so is the leading asymptotics (29). However, for $\varepsilon > 1$ the subleading terms result in oscillations not captured by the leading term, analogously to what happens with the cumulants. This effect is seen in Fig. (6). Note also that for $\varepsilon \neq 1$ the finite system size results in the rapid stop in the linear growth of the entropy with the subsequent saturation once the system's boundaries are reached (*cf.* [62]).

# 7 Loschmidt echo

Let us derive the Fredholm representation (32) of the return amplitude. First assume that the initial state is a pure state $\Psi^{(0)}$, in which case the return amplitude defined in Eq. (31) reads

$$\mathcal{A}(t) \equiv \sum_{\Psi} |\langle \Psi^{(0)} | \Psi \rangle|^2 e^{-it(E_{\Psi} - E_{\Psi^0})}, \tag{112}$$

where $|\Psi\rangle$ ($E_{\Psi}$) and $|\Psi^{(0)}\rangle$ ($E_{\Psi^0}$) are the many-body eigenstates (eigenenergies) of the Hamiltonians $H$ and $H_0$, respectively. The many-body states in Eq. (112) have the form of Slater determinants constructed from the single-particles waves functions, therefore the overlap reads

$$|\langle \Psi^{(0)} | \Psi \rangle|^2 = \left( \det \frac{1}{\cos \varphi - \cos \phi} \right)^2 \prod_{\phi} \frac{1}{2(N+1)g'(\phi)} \prod_{\varphi} \sin^2 \varphi. \tag{113}$$

The Cauchy-like square matrix inside the determinant in the above equation has entries formed by the set $\{\varphi\}$ corresponding to the initial state $|\Psi^{(0)}\rangle$ and the set $\{\phi\}$ (where $\phi$ stands for $\phi^+$ or $\phi^-$) corresponding to the eigenstate $|\Psi\rangle$ (and thus satisfying the spectral equation (38)). The energies read $E_{\Psi} = -\sum_{\phi} \cos \phi$ and $E_{\Psi^0} = -\sum_{\varphi} \cos \varphi$, and the exponent in Eq. (112) factorizes into the product of single particle contributions. As a result we obtain

$$\mathcal{A}(t) = \det \mathcal{M}, \tag{114}$$

with matrix elements defined as

$$\mathcal{M}_{ab} = \frac{\sin \varphi_a \sin \varphi_b}{2(N+1)} \sum_{\phi} \frac{e^{-it \cos \phi}}{g'(\phi)(\cos \varphi_a - \cos \phi)(\cos \varphi_b - \cos \phi)} e^{it(\cos \varphi_a + \cos \varphi_b)/2}. \tag{115}$$

Here we emphasize again that summation is over all solutions of Eq. (38). Taking the sum for $a \neq b$ we get

$$\mathcal{M}_{ab} = -(1+\varepsilon^2) \frac{\sin \varphi_a \sin \varphi_b}{2(N+1)} \frac{S(\varphi_a) - S(\varphi_b)}{\cos \varphi_a - \cos \varphi_b} e^{it(\cos \varphi_a + \cos \varphi_b)/2}, \tag{116}$$

with $S(\varphi)$ defined Eq. (62). The diagonal components $\mathcal{M}_{aa}$ can be computed with the complex analysis trick described in Sec. 3 (see Eq. (51)). This time, however, one has to additionally take into account the residue from the *double* pole $\varphi = \varphi_a$. The corresponding residue is easy to compute and the result reads

$$\mathcal{M}_{aa} = 1 + (1+\varepsilon^2) \frac{\sin \varphi_a}{2(N+1)} S'(\varphi_a) e^{it \cos \varphi_a}. \tag{117}$$

Taking into account spacing between two consecutive $\varphi_a$ (see Eq. (43)), performing transformation to the "energy" space $\varphi \to E = -\cos \varphi$ and accounting for the finite temperature in the same way as in Sec. 3, we represent the return amplitude in the thermodynamic limit as the Fredholm determinant (32) with the kernel (33).

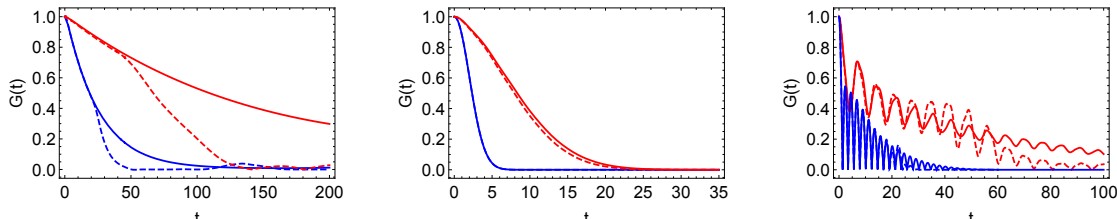

Figure 7: The Loschmidt echo $\mathcal{L}(t) \equiv |\mathcal{A}(t)|^2$ in the thermodynamic limit calculated using the Fredholm determinant representation (32) (solid lines). The blue (red) color corresponds to the full initial filling (initial particle density $\Phi/\pi = 1/3$) of the left. The left, middle and right panels correspond to $\varepsilon = 0.3$, $\varepsilon = 1.0$ and $\varepsilon = 2.4$, respectively. Dashed lines corresponds to the Loschmidt echo in finite system with $N_f = 10$ for the full filling and $N_f = 6$ for the filling one-third.

The leading asymptotics (34) can be obtained performing manipulations similar to those in Sec. (4). First, we approximate the function $S(E)$ according to Eqs. (74), (75) and present the kernel $V$ as the generalized sine-kernel [31]

$$V \approx V_H + \theta(\varepsilon^2 - 1)(\delta V_+ + \delta V_-), \tag{118}$$

which in notations of the previous sections reads

$$V_H(E, E') = \frac{\sqrt{\nu(E)\nu(E')}}{2\pi i} \frac{e^{it(E-E')/2} \mathcal{Z}^*(E) - e^{it(E'-E)/2} \mathcal{Z}^*(E')}{E - E'}, \tag{119}$$

where

$$\delta V_\pm(E, E') = \frac{e_-(E)e_-(E')(\varepsilon - 1/\varepsilon)e^{\mp it\mathcal{E}}}{2(\mathcal{E} \pm E)(\mathcal{E} \pm E')}, \tag{120}$$

and $\mathcal{Z}(E)$ and $\kappa_\varepsilon$ are defined in Eqs. (94) and (67), respectively. The terms $\delta V_\pm$ are present only when $\varepsilon > 1$, but even in this case they do not affect the leading asymptotics. Applying the result of Ref. [31] to the kernel $V_H$ we obtain the leading asymptotics (34).

A remarkable cross-check can be performed for the full initial filling of the left chain. In this case the Loschmidt echo coincides with the probability that none of the particles travelled to the right part of the system. The latter can be expressed through FCS:

$$|\mathcal{A}(t)|^2 = \lim_{\lambda \to -\infty} \chi(\lambda). \tag{121}$$

Due to the identity

$$1 - T(E) = \left(1 - (1 + \varepsilon^2)\mathcal{Z}^*(E)\nu(E)\right)\left(1 - (1 + \varepsilon^2)\mathcal{Z}(E)\nu(E)\right), \tag{122}$$

the leading asymptotics (34) and (18) satisfy relation (121).

In the case $\varepsilon = 1$ the approximation $V \approx V_H$ is too crude, and we were not able to obtain the asymptotics in the case of a general filling. However, in the fully-filled case $\Phi = \pi$, the return amplitude can be computed exactly [36]. Indeed, using a spatial basis one can present the return amplitude as a Toeplitz+Hankel determinant, which can be later evaluated with the help of the Szegö formula [36],

$$\mathcal{A}(t) = \lim_{N \to \infty} \det_{1 \le i, j \le N} (J_{i-j}(t) - J_{i+j}(t)) = e^{-t^2/8}. \tag{123}$$

Let us also add a remark on the case with periodic boundary conditions, i.e. when the "left" and "right" chains are connected by two junctions to form a ring. In this case the return amplitude is expected to be the square of the amplitude for open boundary conditions studied above. This again can be shown with the application of the strong Szegö formula [36, 73],

$$\mathcal{A}_{\mathrm{pbc}}(t) = \lim_{N \to \infty} \det_{1 \le i,j \le N} J_{i-j}(t) = e^{-t^2/4} = \mathcal{A}(t)^2. \tag{124}$$

A comment on the analysis of Ref. [36] is in order. The derivation of the Toeplitz determinant representation (123) in Ref. [36] is not entirely mathematically rigorous, as in fact the corresponding matrix elements turn into Bessel functions only in the limit $N \to \infty$. This motivated us to rederive the result (123) from our Fredholm determinant representation (32) where the limit $N \to \infty$ is already taken. In appendix A.3 we show how to rewrite our kernel $W$ as the Bessel kernel [74, 75] with the result

$$\mathcal{A}(t) = \det_{[0,t]} \left( 1 + \frac{x}{2} \frac{J_0(x)J_1(y) - J_0(y)J_1(x)}{x - y} \right). \tag{125}$$

The Bessel kernel is a particular example of the hypergeometric kernel [76]. Namely, the kernel in Eq. (125) coincides with the expression from Proposition 8.13 in Ref. [76], if one substitutes $r = 1/2$ there and employs the following identity

$${}_1F_1(\nu + 1/2; 1 + 2\nu; 2iz) = (2/z)^\nu \Gamma(1 + \nu) e^{iz} J_\nu(z). \tag{126}$$

Moreover, from Proposition 8.14 [76] it follows that $\sigma(t) \equiv t \, (d\mathcal{A}/dt)$ satisfies the $\sigma$-version of Painlevé V equation

$$(t\sigma'')^2 + (2(t\sigma' - \sigma) + (\sigma')^2)^2 - (\sigma')^2((\sigma')^2 + 1) = 0, \tag{127}$$

which is obviously satisfied for $\sigma = -t^2/4$. Moreover, this solution uniquely specifies the determinant (125) due to the matching of the short-time expansion which serves as the boundary condition for equation (127). This way, we obtain the amazing identity

$$\lim_{N \to \infty} \det_{1 \le i,j \le N} (J_{i-j}(t) - J_{i+j}(t)) = \det_{[0,t]} \left( 1 + \frac{x}{2} \frac{J_0(x)J_1(y) - J_0(y)J_1(x)}{x - y} \right) = e^{-t^2/8}. \tag{128}$$

Providing an elementary proof for this identity seems quite challenging at the moment. Notice also, that for finite $N$ determinant in Eq. (124) is closely related to the partition function of the Gross-Witten-Wadia Unitary Matrix Model [77, 78] (the anti-symmetric model in terms of Ref. [79]). The latter is described by the Painlevé III equation [79, 80]. More specifically, one can relate the partition function (124) with the largest eigenvalue distribution in Gaussian unitary ensemble [81, 82]

$$\mathcal{A}_{\mathrm{pbc}}(t) = e^{-t^2/4} \exp\left( -\int_0^{t^2} ds \, \frac{\sigma_N(s)}{s} \right), \tag{129}$$

with $\sigma_N(t)$ satisfying

$$(s\sigma_N'')^2 + \sigma_N'(\sigma_N - s\sigma_N')(4\sigma_N' + 1) - N^2(\sigma_N')^2 = 0, \tag{130}$$

which means that for $N \to \infty$ only trivial solution $\sigma_N = 0$ is possible, resulting in the final result (124).

In Fig. (7) we show the behavior of the absolute value of the return amplitude (the Loschmidt echo) for different regimes. Note that unlike the current and the entropy behavior

the finite size results coincide pretty neatly with the asymptotic values showing essentially no recurrences. Another observation is that for $\varepsilon > 1$ the presence of the bound state produces oscillations in the echo. Notice that contrary to the FCS the frequency of these oscillations is not determined by the the bound state energy solely but also depends on the filling. This is due to the fact that the asymptotics of the return amplitude (34), contrary to the asymptotics of FCS (18), has non-zero imaginary part which interferes with the finite-rank contributions $\delta V_{\pm}$.

# 8 Discussion

In this paper we studied the non-equilibrium evolution of domain-wall type initial conditions in the free-fermionic one-dimensional system. Our main result is a representation of the full counting statistics and the return amplitude in the form of Fredholm determinants with integrable kernels. The results are valid for thermodynamically large systems and for arbitrary times. At large time numerous asymptotic formulas for Fredholm determinants are available by formulating the corresponding matrix Riemann–Hilbert problem and solving it asymptotically [31, 54, 83]. The heuristic application of these ideas allowed us to extract the leading (hydrodynamic) asymptotics of the full counting statistics, current, shot noise and return amplitude. However, this approximation turned out to be too crude to capture the interesting physics related to the presence of the bound state. The latter is responsible for the oscillating behavior of physical quantities. In particular, the current through the defect experiences persistent oscillations despite the voltage bias between leads (difference of chemical potentials) being constant [38]. We perform a more delicate asymptotic analysis and find that a similar behavior holds for the shot noise rate and the entanglement entropy. The frequency of oscillations equals the energy difference between two bound states localized at the defect. The oscillations of the return amplitude (and of the Loschmidt echo) are observed as well, however, their frequency is not determined solely by the energy of the localized state, but also depends on the initial state. Next-to-leading asymptotic terms are also important in the case of absence of a defect. This case is special in many respects, in particular, by the logarithmic (as opposed to linear) growth of the entanglement entropy at zero temperature (see Eq. (30)). The prefactor of the latter logarithm depends on the initial state. We expect that further understanding of these delicate issues can be gained by means of microscopic bosonization and addressing the corresponding Riemann-Hilbert problems.

We believe that our approach provides useful insights and essential prerequisites for the solution of the interacting case, and at the very least can serve for benchmarking and comparison. We remark that the Levitov-Lesovik type of asymptotic was already implemented in the generalized hydrodynamic approach to interacting systems [24]. Another interesting question our research can be related to is the exploration of dynamical defects (quantum impurities) and moving defects [84–87].

Finally, let us say that the setup theoretically studied here seems to be well in the reach of state-of-the-art experiments with ultracold gases. Ingredients required for such experimental demonstration, including the preparation of an inhomogeneous initial state in a one-dimensional trap and the tunability of the individual lattice hopping were already realised and employed in experimental studies of quantum many-body dynamics. In particular, dynamics of a one-dimensional gas prepared in a spatially inhomogeneous initial state was studied in [88], dynamically tunable optical lattices were used to demonstrate quantized transport in [89, 90], optical lattices with individually tunable lattice sites were realized in [91], and transport through a constriction was demonstrated in [92, 93] (see also a review [94] and references therein).

# Acknowledgements

We thank O. Lisovyy, K. Bidzhiev, A. Bastianello and V. Alba for their comments. O. G. and J.-S. C. acknowledge the support from the European Research Council under ERC Advanced grant 743032 DYNAMINT. The work of O.L. (calculations for finite-size systems) was supported by the Russian Science Foundation under the grant N$^\circ$ 17-12-01587.

# A  Bessel function presentation

## A.1  General considerations

In this appendix we derive representation in terms of Bessel functions for the main expressions. We start with Eq. (54) and consider $\varepsilon < 1$ case. To evaluate this expression we note that

$$\frac{1}{1+\varepsilon^2-2\varepsilon\cos\phi} = \frac{1}{\varepsilon}\sum_{n=1}^{\infty}\varepsilon^n\frac{\sin n\phi}{\sin\phi}. \tag{131}$$

This geometric series is in fact the generating function of Chebyshev polynomials of the second kind $U_n(\cos\phi) = \sin(n+1)\phi/\sin\phi$. The Chebyshev polynomials of the first kind – $T_n(\cos\phi) = \cos n\phi$ appear in the expansion of the exponential function into the Bessel functions basis. Namely, the exponential in the integrand can be represented as

$$e^{-it\cos\phi} = J_0(t) + 2\sum_{n=1}^{\infty}(-i)^n J_n(t)\cos n\phi. \tag{132}$$

The integral in Eq. (54) can be easily evaluated using some trigonometry and the integral relation[4]

$$\int_{-1}^{1}\frac{\sqrt{1-y^2}U_{n-1}(y)dy}{y-x} = -\pi T_n(x). \tag{133}$$

We finally arrive at the expression

$$F(\varphi,t,\varepsilon) = \sum_{n=0}^{\infty}\varepsilon(-i)^n J_n(t)\frac{\varepsilon^n(1-\varepsilon^2)-2\cos n\varphi+2\varepsilon\cos(n+1)\varphi}{1+\varepsilon^2-2\varepsilon\cos\varphi}. \tag{134}$$

One can check that, remarkably, this expression is valid also for $\varepsilon > 1$.

## A.2  No defect

To compare with the known results in literature and get explicit formulas we consider the special case when the defect is absent i.e. $\varepsilon^2 = 1$, and the left part of the system if fully filled i.e $\Phi = \pi$. The expression (134) simplifies into

$$F(\varphi;\varepsilon) = -\sum_{n=1}^{\infty}(-i)^n(J_n(t)+i\varepsilon J_{n-1}(t))\frac{\sin n\varphi}{\sin\varphi}. \tag{135}$$

Therefore, the kernel in Eq. (56) reads as

$$X_{ab} = \frac{e^\lambda-1}{\pi}\int_{0}^{t}d\tau\sum_{n,m=1}^{\infty}(J_n(\tau)J_{m-1}(\tau)+J_n(\tau)J_{m-1}(\tau))\sin(n\varphi_a)\sin(m\varphi_b). \tag{136}$$

---

[4]https://en.wikipedia.org/wiki/Chebyshev_polynomials

The determinant $\chi(z) \equiv \det(1 - z\hat{X})$ can be understood as a trace in the expansion of the logarithm $\log \chi(z) = -\sum_{p=1}^{\infty} \operatorname{tr} X^p / p$. Each trace is computed in the $L_2([0, \pi])$ space. Taking into account that in this space $\int_0^\pi d\varphi \sin(m\varphi) \sin(n\varphi) = \pi \delta_{nm}/2$, we can rewrite each trace in the space of positive integers

$$\operatorname{tr}_{L_2([0,\pi])} X^p = \operatorname{tr}_{L_2(\mathbb{Z}_{>0})} K^p, \tag{137}$$

where matrix $K$ is the discrete Bessel kernel [95, 96]

$$K_{nm} = \frac{1}{2} \int_0^t d\tau \sum_{n,m=1}^{\infty} (J_n(\tau) J_{m-1}(\tau) + J_n(\tau) J_{m-1}(\tau)) = t \frac{J_n(t) J_{m-1}(t) - J_n(t) J_{m-1}(t)}{2(m-n)}. \tag{138}$$

In Ref. [33] it was proved that the discrete Bessel kernel is also equivalent to the continuous one

$$\det_{\mathbb{Z}_{>0}}(1 - zK) = \det(1 - zK_{\text{Bes}}), \tag{139}$$

with $K_{\text{Bes}}$ given by Eq. (83).

Given this presentation we turn to computation of the current given by Eq. (86). The result reads

$$J(t) = \sum_{m=0}^{\infty} J_m(t) J_{m+1}(t). \tag{140}$$

This expression can be simplified even further using techniques similar to Christoffel-Darboux type identities (see for instance [97]). First we rewrite it identically as

$$J(t) = \sum_{m=0}^{\infty} J_m(t)(m+1) J_{m+1}(t) - \sum_{m=0}^{\infty} m J_m(t) J_{m+1}(t), \tag{141}$$

then we use the recurrence condition for Bessel functions

$$\nu J_\nu(t) = \frac{t}{2} (J_{\nu-1}(t) + J_{\nu+1}(t),) \tag{142}$$

to get

$$\begin{aligned} J(t) &= \frac{t}{2} \left( \sum_{m=0}^{\infty} J_m(t)(J_{m+2}(t) + J_m(t)) - \sum_{m=0}^{\infty} (J_{m+1}(t) + J_{m-1}(t)) J_{m+1}(t) \right) \\ &= \frac{t}{2} (J_0^2(t) + J_1^2(t)). \end{aligned} \tag{143}$$

This result was obtained in a slightly different form (and stated with a misprint) in Ref. [35] and in the above form in the arXiv version of [36]. After integration over time one gets an exact expression for the number of particles in the right chain,

$$\langle N_R(t) \rangle = (t^2 J_0(t)^2 + t^2 J_1(t)^2 - t J_0(t) J_1(t))/2. \tag{144}$$

## A.3 Kernel transformation

In this appendix we derive from the determinant (32) an alternative Fredholm determinant representation of the return amplitude with the kernel acting on $L_2([0, t])$. To this end we use the same methods as in Sec. (3). First, we employ relation

$$\partial_t \left( \frac{S(E) - S(E')}{E - E'} e^{-itE'} \right) = i e^{-itE'} S(E), \tag{145}$$

which leads to

$$W(E, E') = i(1 + \varepsilon^2)\sqrt{v(E)\rho(E)v(E')\rho(E')}\frac{e^{it(E'-E)/2}}{2\pi}\int_0^t d\tau e^{-i\tau E'}S(E, \tau). \tag{146}$$

The simplification of this kernel can be obtained by the conjugation with the diagonal matrices without changing the determinant, resulting in

$$W(E, E') = i(1 + \varepsilon^2)\int_0^t \frac{d\tau}{2\pi}S(E, \tau)v(E')\rho(E')e^{-i\tau E'}. \tag{147}$$

Then taking into account that under determinant we can use cyclic property $\det(1 + AB) = \det(1 + BA)$, we can rewrite the Fredholm determinant as

$$\det(1 - W)\Big|_{L_2([-1,1])} = \det(1 + K)\Big|_{L_2([0,t])}, \tag{148}$$

with

$$K(\tau_1, \tau_2) = -i(1 + \varepsilon^2)\int_{-1}^1 \frac{dE}{2\pi}S(E, \tau_1)v(E)\rho(E)e^{-i\tau_2 E}. \tag{149}$$

We compute this kernel exactly when the defect is absent $\varepsilon = 1$ and the initial state is fully filled. Using definition (58) and Eq. (135) one can rewrite kernel as

$$K(\tau_1, \tau_2) = 2i\int_0^\pi \frac{d\varphi}{2\pi}\sum_{n=1}^\infty J_n(\tau_1)\sin(n\varphi)\sin(\varphi)e^{i\tau_2\cos\varphi}. \tag{150}$$

Then using expansion (132) we perform integration and arrive at the following expression

$$K(\tau_1, \tau_2) = \frac{1}{2}\sum_{n=0}^\infty (J_{n-1}(\tau_2) + J_{n+1}(\tau_2))J_n(\tau_1) = \tau_2\sum_{n=0}^\infty nJ_n(\tau_2)J_n(\tau_1). \tag{151}$$

The last sum is a partial case of a more generic Bessel's function analog of the Christoffel-Darboux identity for orthonormal polynomials:

$$\sum_{k=1}^\infty 2(\nu + k)J_{\nu+k}(w)J_{\nu+k}(z) = \frac{wz}{w - z}(J_{\nu+1}(w)J_\nu(z) - J_\nu(w)J_{\nu+1}(z)). \tag{152}$$

Putting $\nu = 0$ in this expression we arrive at the kernel (125).

Using the same technique one can rewrite kernel for the FCS in the block Fredholm determinant form. Again we demonstrate how it works for $\varepsilon = 1$ and $\Phi = \pi$ case. The kernel $X$ in (57) can be rewritten as

$$X_{ab} = \int_0^t \frac{d\tau}{\pi}Z(\varphi_a, \tau).\tilde{Z}(\varphi_b, \tau), \tag{153}$$

where

$$Z(\varphi, \tau) = \left(\sum_{n=1}^\infty J_n(\tau)\sin(n\varphi)\sin\varphi, \sum_{n=1}^\infty J_{n-1}(\tau)\sin(n\varphi)\sin\varphi\right)^{\mathrm{T}}, \qquad \tilde{Z} = \sigma_x Z. \tag{154}$$

Using cyclic permutations we obtain

$$\det(1 + (e^\lambda - 1)\hat{X})\Big|_{L_2([0,\pi])} = \det\left(1 + \frac{e^\lambda - 1}{4}\tilde{K}\right)\Big|_{L_2([0,t])}, \tag{155}$$

where the block kernel $\tilde{K}$ has the following elements

$$\tilde{K}_{11}(\tau_1, \tau_2) = \tilde{K}_{22}(\tau_2, \tau_1) = 2\sum_{n=1}^{\infty} J_n(\tau_1)J_{n-1}(\tau_2), \tag{156}$$

$$\tilde{K}_{12}(\tau_1, \tau_2) = 2\sum_{n=1}^{\infty} J_n(\tau_1)J_n(\tau_2), \qquad \tilde{K}_{21}(\tau_1, \tau_2) = 2\sum_{n=0}^{\infty} J_n(\tau_1)J_n(\tau_2). \tag{157}$$

To evaluate these sums one can use the addition theorems for Bessel function, in particular, we employ the following

$$\sum_{m=1}^{\infty} J_m(x)J_m(y) = \frac{1}{2}\left(J_0(x-y) - J_0(x)J_0(y)\right). \tag{158}$$

Further, one can differentiate this expression over $y$ and use relation $J_n'(y) = J_{n-1}(y) - n\dfrac{J_n(y)}{y}$ along with Eq. (152) to obtain

$$\sum_{m=0}^{\infty} J_{m+1}(x)J_m(y) = \frac{1}{2}\left(\frac{xJ_1(x)J_0(y) - yJ_1(y)J_0(x)}{x-y} + J_1(x-y)\right). \tag{159}$$

This way, the kernel $\tilde{K}$ can be presented as

$$\begin{aligned}
\tilde{K}(\tau_1, \tau_2) = {} & \frac{\tau_1 J_1(\tau_1)J_0(\tau_2) - \tau_1 J_2(\tau_2)J_0(\tau_1)}{\tau_1 - \tau_2}\mathbf{1}_2 \\
& + J_1(\tau_1 - \tau_2)\sigma_z + J_0(\tau_1 - \tau_2)\sigma_x - i\sigma_y J_0(\tau_1)J_0(\tau_2).
\end{aligned} \tag{160}$$

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
