# Peer review of "Fredholm determinants, full counting statistics and Loschmidt echo for domain wall profiles in one-dimensional free fermionic chains"

_SciPost Physics, doi:SciPost Phys. 8, 036 (2020)_

## Round 1 · Referee Report · Anonymous · 2020-1-6

Report

The authors report on the exact computation of the time evolution of various nonequilibrium observables of interest, such as current, shot noise, return amplitude and entanglement entropy, in a chain of hopping fermionic particles, starting from a quite general imbalanced configuration. The topic is timely due to active theoretical and experimental interest in the dynamics of inhomogeneous systems with conservation laws. The main manuscript contribution is a technical innovation in the computation of the full counting statistics of transported charge via Fredholm determinants. Also, the physical effects of a defect at the location of the initial “domain wall” are studied in detail.

Although the paper is extremely technical, the physical content is clearly explained and accessible to a nonspecialist. This study is a useful addition to the literature and the authors perform an excellent job in comparing with previously known results and indicating the original contributions of this study. Calculations are properly presented (I would make the derivation of Eq. (45) more pedagogical, as it is a central point). I have essentially nothing to criticize about this manuscript, and I think it is suitable for publication in SciPost in the present form.

As a side remark, the authors might find it interesting to compare the a.c. current induced by the defect, which they discuss in this manuscript, with that induced by the superficially similar (but seemingly unrelated) effect studied in P.P. Mazza et al., Phys. Rev. B 99, 180302(R).

---

## Round 1 · Referee Report · Anonymous · 2020-1-30

Strengths

1-Calculation clearly presented and easy to follow
2-Comprehensive and elegant exposition of many results scattered in the literature

Weaknesses

1-Some of the results are not new

Report

The authors revisited the calculation of the particle Full Counting Statistics (FCS), the half chain Entanglement Entropy (EE) and the Loschmidt Echo (LE) for a systems of free fermions with a weak link (parameterized by $0<\varepsilon<1$). In the initial state the right half is empty, while the left half is characterized by temperature, and density (or chemical potential at finite temperature).

The authors derive Fredholm determinant representations for the FCS and the LE, from which in most of the cases they are able to extract exact large-time asymptotic.

Although it is true that most of the results presented here already appeared in some form elsewhere, I found the Fredholm determinant derivation of the FCS and the LE comprehensive, elegant and pedagogically written.
The bound state oscillatory contributions to the current and the entanglement are also clearly emphasized.

I believe that the results presented are ready for publications in their present form.

I spot a few innocent typos, listed below:

-Eq. (7) differential is missing
-Eq. (42), $\varphi\rightarrow\phi$
-Eq. (66), $t\rightarrow\tau$
-Below Eq. (86), I think $N\rightarrow N_f$

Requested changes

See report

---

## Round 2 · Author Response

We thank referees for careful reading of our manuscript. As requested we have corrected the typos and added references.

---

## Round 2 · List of Changes

Eqs. (7), (42), (66) and text below (86) are corrected. References [48] - [52] added.

You are currently on this page

Resubmission 1911.01926v2 on 4 February 2020

---

## Editorial Decision

published